# THE CHALLENGING GROWTH:
# EVALUATING THE SCALABILITY OF CAUSAL MODELS

## ABSTRACT

One of the pillars of causality is the study of causal models and understanding under which hypotheses we can guarantee their ability to grasp causal information and to leverage it for making inferences. Real causal phenomena, however, may involve drastically different settings such as high dimensionality, causal insufficiency, and nonlinearities, which can be in stark contrast with the initial assumptions made by most models. Additionally, providing fair benchmarks under such conditions presents challenges due to the lack of realistic data where the true *data generating process* is known. Consequently, most analyses converge towards either small and synthetic toy examples or theoretical analyses, while empirical evidence is limited. In this work, we present in-depth experimental results on two large datasets modeling a real manufacturing scenario. We show the nontrivial behavior of a well-understood manufacturing process, simulated using a physics-based simulator built and validated by domain experts. We demonstrate the inadequacy of many state-of-the-art models and analyze the wide differences in their performance and tractability, both in terms of runtime and memory complexity. We observe that a wide range of causal models are computationally prohibitive for certain tasks, whereas others lack in expressiveness. We release all artefacts to serve as reference for future research on real world applications of causality, including a general web-page and a leader-board for benchmarking.

## 1 INTRODUCTION

The mastery of *Causal Reasoning* is a long-standing challenge in AI, with the potential to drastically impact many disciplines including medicine, science, engineering, and social sciences. The development of agents with an understanding of causality enables them to go beyond statistical co-occurrences, and is connected with desirable abilities such as reasoning and Out-of-Distribution generalization (Richens & Everitt, 2024). Using the tools of *Causality* (Pearl, 2009) we can uncover the *Data Generating Process* (DGP), and manipulate it to gain a better understanding of the system being modeled. With *Causal Inference* we can estimate the effect of interventions on a system while accounting, among others, for confounding biases and missing data (Mohan & Pearl, 2019). To make progress in this area, a fair and comprehensive evaluation of causal algorithms is crucial, as well as benchmark tests analyzing methods from different angles. Laying down a comparison across multiple domains, however, presents various challenges. From a practical perspective, one of the main obstacles that impedes progress in causality is the lack of public benchmarks supporting method evaluation (Cheng et al., 2022). When benchmarking on real world data, the true DGP may be partially or even completely unknown. Additionally, an individual can either be treated or not, which means that we cannot simultaneously observe both potential outcomes, implying that the ground truth values of the causal estimands are not known. Consequently, purely factual observational data is insufficient for evaluation due to the unavailability of counterfactual measurements. A similar challenge is indicated by Gentzel et al. (2019), who stressed the importance of evaluating on interventional measures and downstream tasks. In most cases, however, obtaining interventional data is not possible, unethical, or highly expensive. Shifting to simulated data, Curth et al. (2021) argued that algorithms matching the assumptions of the DGP are advantaged in those specific benchmarks, but results may not transfer to other scenarios. Despite this, when correctly designed, simulation can be a powerful tool to benchmark causal models. Thanks to causally-plausible simulators, we can obtain any interventional distribution while retaining control on every parameter knob, with the

possibility to study any valuable corner case. Along this path, we can use simulations to gain insights on the behaviour of causal models at the intersection of non-linearity, causal in-sufficiency and high dimensionality. For the latter, bringing causality to the large scale has been the main driver for a series of efforts (Tigas et al., 2022) that tried to understand the scalability issues that several causal models have when dealing with thousands of variables, as well as their inference limitations when performed with finite resources. Scalability is a challenge not only for inference tasks, but also throughout the whole field of causality. The related task of *Causal Discovery* (CD) i.e., recovering the causal diagram from data, suffers from similar burdens, where often mathematical guarantees are sacrificed in exchange of computational feasibility (Zheng et al., 2018b). Hereby, we investigate how those methods perform at large scale, and consequently aim to answer the question whether current approaches are adequate for realistic scenarios. Our doubt stems from the looming intractability that current methods possess *by design* (Eiter & Lukasiewicz, 2002) when carrying out certain tasks, both from a theoretical and practical viewpoint. Furthermore, we try to motivate the statement that mathematically sound large-scale causality may require new methodologies and engineering breakthroughs that are not yet developed.

**Contributions** The present paper fills the gap between small controlled benchmarks from one side, and real world (but hard to evaluate on) scenarios on the other. Novel causal models are often tested on representative causal graphs (chain, napkin, etc.) with simple structural equations, which lack the complexity of the real world. Differently from other works which explore applications of causality to medicine, genetics and ecology, we focus on the manufacturing domain, which has found only experimental and scattered applications in the past (Vukovic & Thalmann, 2022; Göbler et al., 2024)

Specifically, our contributions are three-fold:

- We perform various case studies on the capabilities and limitations of a diverse range of causal models. To sustain our analysis, we work on complex and realistic datasets generated with a simulator based on physical models derived from first principles and expert knowledge. We investigate these models at large scale on exemplary tasks at the interventional level with the goal of highlighting their differences in terms of performance and tractability (time and memory-wise).
- We execute similar analyses for Causal Discovery, comparing classic algorithms and recent learning-based methods.
- We release the two large size benchmark datasets on the manufacturing domain, on which our experiments are performed, with the aim of fostering research in high dimensional causality. Each dataset comprise over a million of samples, including both observational and interventional data sampled from two Structural Causal Models. Additionally, we release the DGPs, enabling researchers to generate new observational and interventional data.

## 2 RELATED WORK

In this section we analyze related approaches relevant to our work and datasets, highlighting common points and dissimilarities. For more exhaustive surveys on the evaluation of causal models, we address the interested reader to Cheng et al. (2022), Guo et al. (2021) and Yao et al. (2021).

**Large-Scale Causality:** In Zečević et al. (2023), a theoretical and empirical evaluation on simple causal graphs highlighted the intractability of marginal inference and the scaling laws of different causal models. When the goal is to reduce the complexity of different intractable queries, it is possible to adopt *tractable probabilistic models* such as *Sum-Product Networks* (SPNs) (Poon & Domingos, 2012). Furthermore, it is possible to use SPNs to model causal phenomena(Zecevic et al., 2021; Busch et al., 2023; Poonia et al., 2024; Busch et al., 2024).

Leveraging its independence from combinatorial objects such as graphs, *Rubin's Potential Outcomes* (PO) framework (Imbens & Rubin, 2015) can be used to tackle the scalability problem. However, a notable limitation of the PO framework is its reliance on assumptions like *ignorability*, that is equivalent to unconfoundedness and is not suitable for our strongly confounded use-case.

In the realm of causal discovery, scaling is addressed with novel methodologies such as continuous optimization-based approaches (Zheng et al., 2018c; Ng et al., 2020; Lachapelle et al., 2020) or divide-and-conquer approaches (Lopez et al., 2022; Wu et al., 2024). However, while easier to

scale, they suffer from distinct vulnerabilities. Reisach et al. (2021) and Kaiser & Sipos (2021) show that their performance is sensitive to the scale of the data, and can degrade to levels comparable to or worse than classic approaches after data normalization. On a similar note Loh & Bühlmann (2014) and Seng et al. (2024) remarked the limitations of methods relying on mean squared error losses. Further, Mamaghan et al. (2024) studied the drawbacks of common metrics when adopting a Bayesian approach. Those drawbacks of ML-Based approaches re-ignited interest in novel and more mathematically grounded methods such as *Extremely Greedy Equivalence Search* (XGES) Nazaret & Blei (2024) or *Differential Adjacency Test* (DAT) Amin & Wilson (2024).

**Datasets and Benchmarks:** A wide variety of benchmarks for causal models are publicly available (Lauritzen & Spiegelhalter, 1988; Beinlich et al., 1989; Sachs et al., 2005). However, only a limited number of them target large scale scenarios Andreassen et al. (1991), and an even smaller fraction involve hybrid domains, which is the focus of our datasets and experiments. To compensate the lack of data, a common choice for analysing scaling laws for causal models is to generate random Erdos-Renyi (Erdos & Rényi, 1984) or Scale-Free graphs (Barabási & Albert, 1999) which, although easy to simulate, are far from reflecting the real world. Recent works provide datasets and methodologies to generate realistic synthetic and semi-synthetic data. Semi-synthetic DGPs tuned on real data, often along with the use of prior domain knowledge, are the focus of simulators such as *CausalAssembly* (Göbler et al., 2024) for the manufacturing domain, or the *Neuropathic Pain simulator* (Tu et al., 2019) in the medical domain. Further, semi-synthetic DGPs are used in Dorie et al. (2017); Hahn et al. (2019) and Shimoni et al. (2018) to generate datasets with real observational data for the untreated individuals, coupled with simulated treated counterparts. Contrary to those datasets, our data comprise additional layers of complexity by simulating mechanisms such as batching, hybrid data-types and conditional dependencies. Concentrating on real world data, Causal-Bench (Chevalley et al., 2022) is a large scale benchmark for single-cell perturbation experiments with interventional data gathered using gene-editing technologies. A different strategy is adopted by CausalChambers (Gamella et al., 2024), which builds a real isolated physical system where physical mechanisms are known almost perfectly, giving a high degree of confidence on the exactness of the ground-truth Structural Causal Model. Additionally, Mogensen et al. (2024); Mhalla et al. (2020) provide real-world datasets with a more or less justified ground-truth causal graph.

## 3 BACKGROUND

### 3.1 CAUSAL MODELS

Modern causality in the Pearl sense relies on intuitive graphical representations of causal phenomena. Here, we assume that the underlying causal structure can be represented by a *Directed Acyclic Graph* (DAG) $\mathcal{G} = (E, V)$, where the sets $V = \{1, \ldots, d\}$ and $E \subseteq V \times V$ are vertices and directed edges respectively. Direct causes of a node $v_i$ are called Parents and are denoted with $Pa_{\mathcal{G}}(v_i)$.

We start by defining *Structural Causal Models*, which incarnate the Pearlian notion of causality (Pearl, 2009) and defines the DGP.

**Definition 1.** *A Structural Causal Model (SCM) is a 4-tuple $\mathcal{M} := (\boldsymbol{U}, \boldsymbol{V}, P_{\boldsymbol{U}}, \mathcal{F})$ where*

- $\boldsymbol{U}$ *is the set of exogenous variables that are related to external factors,*

- $\boldsymbol{V}$ *is the set of endogenous variables that depend on other endogenous/exogenous ones,*

- $P_{\boldsymbol{U}}$ *is the probability density function of the exogenous variable $\boldsymbol{U}$,*

- $\mathcal{F} = \{f_1, f_2, \ldots, f_n\}$ *is the set of Structural Equations, where each element is a mapping such that $f_i : U_i \cup Pa_i \to V_i$, with $U_i \subseteq \boldsymbol{U}$ and $V_i \subseteq \boldsymbol{V}$. Each endogenous variable is related to a structural equation that determines its values. In practice, each node $v_i \in V$ can be expressed as $v_i = f_i(u_i, Pa_i)$.*

Looking at the dependency structure between variables induced by Structural Equations it is possible to derive a causal graph for the phenomena being modeled. Furthermore, when we assume that the dependency on exogenous variables is additive in the form $v_i = f_i(Pa_i) + u_i$, we say that the SCM adopts an *Additive Noise Model* (ANM).

Causal models can be classified in 3 Layers or rungs, namely the *Pearl Causal Hierarchy* (Bareinboim et al., 2022), where a Model in the second layers is called *Interventional* if it can model Interventions (manipulations of the causal structure), and *Counterfactual* if it can model Counterfactuals (*what-if* queries). Our focus will be on layer 2, with the goal of making estimates for different interventional queries. In section 5 we show how different causal models may have radically different properties and computational requirements for the same causal query.

Lastly, even though the complete description of the causal phenomenon is assumed to be a DAG, its marginalisations to lower dimensions may not be DAGs. Indeed, if a set of variables is marked as latent, the operation of marginalizing out latent variables is called *latent projection* (Verma & Pearl, 2013), which can result in a graph containing directed but also bi-directed edges representing causal relationships confounded by a latent variable, called *Acyclic Directed Mixed Graph* (ADMG).

## 3.2 TREATMENT EFFECT ESTIMATION

The most common tasks in *Causal Inference* (CI) involves the prediction of the effect of one or multiple interventions on an outcome variable and assess its effectiveness i.e., the *Treatment Effect*. Treatment Effect estimation is based on comparing a population of treated individuals with a reference control group that did not receive any treatment.
We proceed by defining the *Average Treatment Effect* (ATE) which describes how, on average, an individual responds to a specific treatment:

$$ATE = \mathbb{E}[Y(1) - Y(0)],$$ (1)

where $Y(1)$ and $Y(0)$ indicate respectively the outcomes in presence or absence of a treatment.
When Searching for fine-grained estimates, we can encounter scenarios where treatments will affect different sub-populations heterogeneously e.g. *Heterogeneous Treatment Estimation* (HTE). To identify the treatment effect to such level of detail, we condition the ATE on $X = x$, and define the *Conditional Average Treatment Effect* (CATE) as

$$\tau(x) = \mathbb{E}[Y(1) - Y(0)|X = x].$$ (2)

We note that ATE and CATE estimates rely mostly on comparing treated and untreated individuals. This brings us to the *Fundamental Problem of Causal Inference* (Rubin, 1974), which states that an individual can either be treated or not. Consequently, $Y(1)$ and $Y(0)$ are never observed simultaneously and can only be estimated.

## 4 CAUSALMAN: THE MANUFACTURING DATASET

This simulator is based on physical models derived from first principles (described in C.2), and two large-size SCM are provided. To provide the most realistic environment, domain experts have been heavily involved during the entire workflow, including the validation/fine-tuning of simulation hyper-parameters, and the definition of all physical models (e.g. structural equations) involved in the production life-cycle. Additionally, we simulate dedicated mechanisms specific to production lines such as *Batching*, which also influences the sampling process. In this simulated environment, we can generate unlimited observational and interventional data, including accurate estimates for any ground truth ATE and CATE. Table 1 provides an overview on the scale of our datasets, both in terms of dimensionality and number of samples.

**Hydraulic Units, Blocks and Magnetic Valves**: We are modeling an assembly line that assembles *Hydraulic Units* (HU). An HU is a device used to control the flow of a fluid. It is composed by an *Hydraulic Block* (HB) and by a certain number of *Magnetic Valves* (2 for CausalMan Small and 8 for CausalMan Medium). An HB is a mechanical component with a different number of bores where, during the assembly process, MVs are inserted into them with a press-fitting machine. A *Magnetic Valve* (MV) is the electromechanical component inside the HU thanks to which, after applying a voltage, it is possible to control the flow of a fluid. In practice, by energizing the MVs we can control whether the fluid can flow or not through the HU. The faults that we are modeling are related to the leakage of fluid through the MV and through the HU in situations where it is not supposed to happen. Those faults are often caused by anomalies during the Press-Fitting (PF) process, or can be caused by some material properties of the MV or HB not being ideal. Further details in C.2.

| Dataset | Full Graph | | Observable Graph | | # Samples | |
|---------|------------|---|------------------|---|-----------|---|
| | Nodes | Edges | Nodes | Edges | Obs. | Int. |
| Small | 157 | 121 | 53 | 95(13) | 717.962 | 622.385 |
| Medium | 605 | 1014 | 186 | 381(172) | 717.911 | 620.537 |

Table 1: Overview of the two datasets. On the left column we list the information for the full causal graph, while on the right for the partially observable graph. In parentheses we have the number of bi-directed edges. All our experiments use the partially observable (therefore causal insufficient) causal graph.

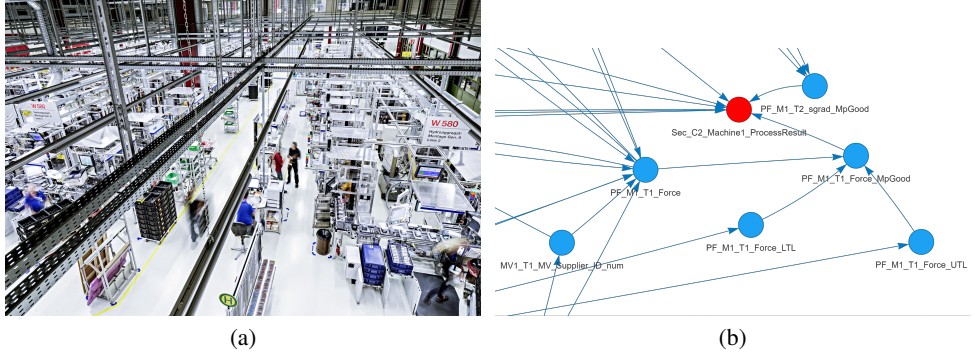

| (a) | (b) |
|-----|-----|

Figure 1: In Fig. 1a, a photo of the real production plant being simulated. In Fig. 1b, a subgraph of the ground truth causal graph for both datasets. In our treatment effect estimation tasks, "Sec_C2_Machine1_ProcessResult" is the outcome binary variable, whereas interventions will be applied on other binary and discrete target variables. Further information in Sec. 4.1.

## 4.1 DATA DESCRIPTION

In this section we describe the most important aspects of our datasets. We acknowledge that, given the complexity of the DGP used to generate our datasets, most assumptions on which the vast majority of causal models rely are not fulfilled. Therefore, identifiability is likely to not hold anymore.

**Data-Types:** Regarding the domain of the covariates, our datasets exhibit Mixed Data-types with Continuous, Booleans and Categorical variables. In F we describe how the data is pre-processed and numerically embedded before running our experiments.

**Structural Equations and noise models:** Each structural equation is defined by relying on prior domain knowledge. Moreover, also hyper-parameters related to source node distributions are defined by domain experts with the intent to mirror the real world production line. Additionally, dependencies on exogenous variables are often nonlinear and source node distributions can differ between samples (See 4.2 for additional details), hence we are not dealing with any underlying ANM.

**Conditional Dependencies:** Certain node distributions are determined (i.e., caused) by specific combination of categoricals, see Fig.4 in the appendix. Given a node $n_i$ describing an attribute, its node distribution may depend on the value of different categorical parent nodes such as the supplier or the component type. Therefore, by varying the categorical values of a parent, the hyperparameters determining the distribution of $n_i$ can change.

**Causal Insufficiency:** Although the complex physical mechanisms are well-known, in the real system it is possible to measure only part of the variables, therefore every simulated variable has been marked either as observable or hidden by domain experts, with the goal of reflecting as accurately as possible the real system. All our experiments use the ADMG obtained after a latent projection to marginalise out latent variables (See Table 1 for more details).

**Monitoring** Production lines typically incorporate anomaly detection mechanisms for the purpose of identifying faulty parts that are not fit for use. In the best case scenario, a defective product should be caught soon and removed (scrapped) before reaching the end of the production line. Analogously, in our simulated environment many attributes have to stay within specific ranges of values (See Fig.

5 in the appendix). This is modeled with a boolean variable that can be either true or false depending if an attribute is in the correct range or not. Further, the range of values for every attribute is described by a *Lower Tolerance Limit* (LTL) and an *Upper Tolerance Limit* (UTL), which depends (are caused) from the type of component being produced (see Fig. 5 and Sec. C).

$$\text{MpGood}_i = \begin{cases} True & \text{if } \text{LTL}_i \leq x_i \leq \text{UTL}_i, \\ False & \text{otherwise.} \end{cases} \tag{3}$$

At the end of every process, a *logic AND* operation between every *MpGood* (Mechanic-Part Good) variable is performed to check if all the attributes within the machines fall within the desired range. If that is true, the variable ProcessResult, which signals the quality conformance of the final product, will be *True*, otherwise *False*. In the real scenario, if the process result is false, the component is scrapped because at least one of the parameters is not within the acceptable range.

### 4.2 BATCHING AND SAMPLING PROCEDURE

**Batching:** Our simulator replicates an important mechanism typical of real production lines, namely *Batching*. Batching is the subdivision of production in *batches* i.e., groups of parts being produced together and share similar properties. On the same production line there may be different batches producing different products. All those batches share the same causal structure, and within a batch the parametrization is the same, therefore we can perform *ancestral sampling* (Koller & Friedman, 2009) on the SCM *related to the batch*. Although the SCM is constant across batches, individual parametrization can differ, which consists for example in the variation of hyperparameters related to source distributions. In practice, for every batch we set one parametrization of the SCM, and only then we perform ancestral sampling. For next batches we repeat this procedure by setting new parameters on the SCM and then sampling again. This complex sampling procedure generates diverse and rich and heterogeneous datasets. Different products identify distinct sub-populations, providing an ideal playground for testing various HTE techniques.

**Interventional data:** Interventions are defined within a batch, and Interventional data is sampled by first setting the correct SCM parameterization relative to the batch, and then applying the hard/soft intervention. Next, ancestral sampling is performed as for observational data. In other words, we have *Interventional Batches* where a batch is sampled while an intervention is being applied. This procedure is also applied when sampling the ground truth data for treated and control groups during the treatment effect estimation experiments.

## 5 EXPERIMENTS

In this section we list and describe the causal models and causal discovery algorithms of our choice, and the general experimental setting. Additional implementation details are present in F.

### 5.1 CAUSAL MODELS

We perform experiments on a representative set of causal models, with the goal of highlighting the different characteristics that those methods possess by design. We test *Causal Bayesian Networks* (CBN) (Bareinboim et al., 2022), *Neural Causal Models* (NCM) Xia et al. (2022a), Normalizing Flows-based models such as *CAREFL* (Khemakhem et al., 2021) and *Causal Normalizing Flows* (Javaloy et al., 2023), and *Variational Causal Graph Autoencoders* (VACA) (Sanchez-Martin et al., 2021). Lastly, when estimating treatment effects, we also consider regression-based techniques such as *Linear* and *Logistic Regression*. E.2 provides a more detailed description of the chosen models.

### 5.2 CAUSAL DISCOVERY ALGORITHMS

A wide variety of Causal Discovery algorithms are investigated as well. We start from traditional Constraint-based ones, to more recent score-based approaches that involve machine learning. We test classic methods such as the *Peter-Clark* (PC) algorithm (Spirtes et al., 2001), its variant *PC-Stable* (Colombo & Maathuis, 2014), and *Linear Non-Gaussian Additive Noise Models* (LiNGAM) (Shimizu et al., 2006). For learning-based approaches, we test NOTEARS (Zheng et al., 2018a), GOLEM (Ng et al., 2020), DAG-GNN Yu et al. (2019) and GranDAG (Lachapelle et al., 2020).

Additionally we capture metrics for a random *Erdos-Renyi DAG* in every experiment to establish how distant those methods are from random guessing.

## 5.3 CASE STUDIES

We formulate three different case studies. The first two target causal inference tasks using the ground truth ADMG. The third one emulates a real-world scenario where the correct ground truth causal graph is not available, forcing us to perform causal discovery prior to any other task.

**(ATE):** We estimate the ATE for a binary variable indicating the success of the production process, which is 1 if the individual product (sample) is produced correctly and 0 otherwise. As stated in Sec. 4.1, its value depends on multiple binary parents which describe whether different parameters are within the correct range. Therefore, in our *first ATE task* we intervene on one of them, setting it to 0. As a result, the interventional distribution will be 0 with 100% probability. In the *second ATE task*, the treatment is an intervention on a lower tolerance value, which is raised to a higher value, with control value set to a lower one. The target variable is now discrete and not binary anymore, and the target is a grandparent of the outcome variable. As a result of this intervention, the true interventional distribution has a higher probability of being 0 compared to the observational distribution. In practice, the number of samples classified as not good (ProcessResult = False) will increase. Finally, we run a *third ATE task* where we want to understand the effect of increasing the press-fitting force (further information in Sec. B). This variable is connected to the outcome variable through a long path, and extreme values generate a chain of different anomalies in its descendants. Additionally, there are multiple confounded relationships between target and outcome variables.

**(CATE):** Interventions on parameters may have heterogeneous effects across different sub-populations. Consequently, ATE estimates provide a general insight on the behavior of the system, but cannot capture how different sub-populations react to the treatment, which is why in this case study we adopt a more targeted approach by estimating different CATEs. In our dataset, we can think of product types as sub-populations, where interventions on parameters can impact positively the quality of one product while degrading another. Therefore, we repeat the same interventional experiments as in C1 *while conditioning* on a categorical variable (the product type).

**(Discovery):** As a last case-study, we perform Causal Discovery on our datasets. We observe the consistency of methods, and if any of those CD methods can discover a path between target variables and outcome, as the latter is of crucial importance for the CI downstream task. Our goal here is two-fold: 1) we test those CD methods on a realistic scenario with normalized data; and 2) to provide additional empirical evidence on the performance and limitations of ML-Based CD methods, which often offer weaker mathematical guarantees.

### 5.3.1 EVALUATION METRICS

**Metrics for Causal Inference:** In a simulated environment, ground truth interventional quantities are available, therefore we measure the distance between the estimated interventional distributions and the ground truth using the Mean Squared Error (MSE), Jensen-Shannon Divergence (JSD) (Lin, 1991) and Maximum Mean Discrepancy (MMD) (Gretton et al., 2012). For treatment effects, we measure the MSE between the estimated effect and the ground truth obtained from the simulator.

**Metrics for Causal Discovery:** We will measure common metrics such as Structural Hamming Distance (SHD), Structural Intervention Distance (SID) (Peters & Bühlmann, 2014), parent-Separation Distance (p-SD) (Wahl & Runge, 2024), Precision and Recall, as described in E.1.

**Runtime Metrics:** For each causal model, we measure their training/discovery time and their memory usage. For each model that uses GPUs (NCM, CAREFL, CNF, and VACA), we additionally report its average GPU memory usage. Each GPU run was executed on a single A100 GPU. Finally, to capture the general behavior, each experiment is repeated 5 times with different random seeds. In our results we average across the seeds and report mean and SD. for each metric.

| Model | ATE MSE | CATE MSE | JS-Div Tr. | MSE | MMD |
|-------|---------|----------|------------|-----|-----|
| CBN | 1.433(0.061) | 1.653(0.035) | 0.319(0.002) | 0.742 (0.003) | 0.734(0.116) |
| NCM | 1.75(0.068) | 1.502(0.141) | 0.589 (0.000) | 1.000 (0.000) | 0.396(9.023) |
| CAREFL | 1.332 (0.211) | **1.574(0.288)** | 0.512 (0.093) | 0.939 (0.088) | **0.035 (0.087)** |
| CNF | 1.913(0.018) | 1.8(0.04) | **0.291(6e-5)** | 0.707 (0.000) | Nan |
| VACA | 1.907(0.009) | 1.974(0.274) | 0.332(0.01) | **0.339 (0.005)** | 0.319(0.009) |
| Linear r. | **0.229(0.004)** | - | - | - | - |
| Logistic r. | 1.439(0.008) | - | - | - | - |

Table 2: Comparison for the first ATE task on CausalMan Small with $n = 50.000$ samples and ground truth ADMG. Instabilities during sampling prevented to evaluate MMD for CNF, as multiple datapoints diverged to $+\infty$.

.

# 6 RESULTS AND DISCUSSION

### 6.0.1 CAUSAL INFERENCE

**Performance**: Table 2 shows the causal inference performance for the first two case studies. Surprisingly, in the first and simplest task we observe how a simple linear regression is outperforms all other causal models. For regression-based methods, we can explain this result by considering that the intervened variable is on the *markov blanket* of the outcome, making this behavior expected in a SCM-based DGP. We notice that for every causal model, apart from regression-based techniques, ATE or CATE is not estimated directly. Indeed, in those models treatment effects are estimated by averaging over samples from the interventional distributions for treated and control populations. Interestingly, deep causal models exhibit superior performance when estimating the treated interventional distributions while being highly inaccurate for treatment effect estimation (Fig.11). This can be explained by looking at the discrepancy between the JS-Divergence of the reconstructed interventional distributions for the treated and control groups in Figure 2b. It can be clearly seen that, even though the treated population is modeled perfectly, the control population is almost randomly guessed. However, accurate treatment effect estimation using those models require precise reconstructions of both treated and control distributions, and the best-performing models overall are simple regression-based techniques that do not go through this procedure and target ATE or CATE directly. As shown in Table 3, switching to the second treatment estimation task, which is slightly harder, leads to inaccurate results for most models, including regression-based methods. On the third task, which deals with confounded and nonlinear causal mechanisms, the deterioration of linear regression is evident, as it is now the worst-performing method (See Table 4). A similar behavior is present when estimating CATE as well. All causal models indeed fail to reproduce simple conditional interventional distributions. Furthermore, all results transfer to CausalMan Medium.

Therefore, given the poor performance of all causal models for simple Treatment Effect Estimations, what are the advantages of using them? The first and foremost answer comes from the origins of Causality, therefore *robustness against confounders*. Regression-based techniques based on the PO framework often rely on the ignorability assumption, which is identical to unconfoundedness, thus limiting their applicability to phenomena where confounding effects are more prevalent (See third ATE task in Fig. 10 and Table 4). Moreover, modeling directly ATE or CATE is not sufficient in settings where investigations occur on a purely counterfactual level. Indeed, *Explainable AI* techniques may benefit from the counterfactual capacities of these models to build enhanced *causally-coherent* explanations Janzing et al. (2020).

**Computational Scaling:** From a computational perspective, the results reveal an interesting and diverse landscape of model behavior. For CBNs, which are capable of handling only discrete variables, continuous variables have been uniformly quantized in a finite number of steps. However, this design choice is associated with an explosion in memory requirements during the fitting process. This is due to the combination of a high number of states and the in-degree (e.g, parents) of some nodes, which leads to an exponential increase in the number of conditional probability distributions to be estimated. To limit memory requirements and make the computation tractable, we restrict the number of quantization steps to 20, as a higher number would lead CBNs to demand

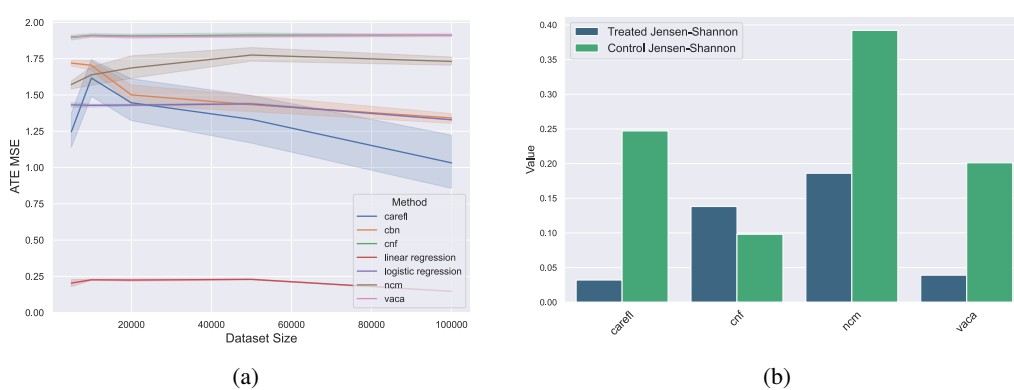

(a)                                                    (b)

Figure 2: CausalMan Small. Figure 2a shows a stagnation in performance for effect estimation, even with the use of more data. Figure 2b, instead, illustrates the JS-Div. accuracy of treated and control distributions for learning-based causal models, after training with $n = 50.000$ samples.

amounts of memory that are impossible to satisfy. No experiments were possible on the second dataset for the same reason, even after aggressively quantizing the training data.

Contrarily, deep models follow different scaling laws, as their complexity is mainly related to the number of parameters in the network, rather than to the number of nodes. In other words, large-scale causality does not directly imply a higher number of parameters, but larger causal graphs may require a higher model capacity to be learned, and consequently bigger neural networks. Among deep models, NCMs are proven to be the most computationally expensive. Figure 8 shows a long runtime and significant memory demands for training, thus limiting possible applications to large-size causal graphs. Due to the significant time required for convergence, it is essential to mantain a high batch size to ensure a reasonable training time. However, there are memory limitations when increasing the batch size, which impose a constraint on the maximum size of the dataset that NCMs can handle. This is a characteristic of the model related to its current training procedure and architecture of each individual parameterized structural equation, as shown in Zečević et al. (2023).

**How much data is actually needed?** Architectural innovations play a crucial role in the success of machine learning (Vaswani et al., 2023; Gu et al., 2022), as they allow to efficiently leverage large amounts of data and compute to improve performance. In Figures 2a and 7a, however, we can see that all models for both CI and CD did not improve significantly with the increase in size of the dataset. In the future, where causal models are applied to datasets with even higher dimensionality, such as multimodal data, it will be crucial to develop scalable models.

### 6.0.2    CAUSAL DISCOVERY

Tables 7 and 8 show results for causal discovery, and Sec. D provides additional results. All algorithms are far from providing an accurate reconstruction of the causal graph in both datasets. Moreover, their SHD performance is almost independent of the dataset size (Figure 7a and 7b), suggesting a limited capacity of leveraging large amounts of data. In CausalMan Small, classic methods such as PC or LiNGAM algorithms remain competitive with ML-Based methods. This is due to the fact that this dataset constitutes an intermediate ground where those methods can still manage the dimensionality of the problem, both performance-wise and resource-wise. In contrast, when scaling to CausalMan Medium, their limitations are visible in Fig. 6 and 3 where, in the latter, we can see that their runtime is multiplied by 20 to 40 times upon tripling the nodes in the graph. Additionally, Fig. 8 indicates additional limitations, this time with respect to the dataset size, where another significant increase in computation resources occurs. The decreasing performance of the PC algorithm on the second dataset can also be explained by the inapplicability of conditional independence tests on large graphs, as the probability of finding a d-separating set is infinitesimal as the number of variables tends to infinity (Feigenbaum et al., 2023). As causality is scaled to large graphs, the SHD loses its relevance. The reason is that SHD is a global metric that becomes too coarse with large graphs, and that does not take into account the *error distri-*

*bution.* A causal discovery algorithm may provide a perfect reconstruction of one unimportant part of the graph, while missing some edges of crucial importance for the CI tasks of interest.

Therefore, we suggest that a fine-grained metric dependent on the specific CI task is needed. Additionally, SHD is only a *structural* metric that relies on counting wrong edges, and is not directly tied to the causal phenomena under the lens. Our analysis demonstrates that current CD methods, when dealing with large graphs, can only be part of an exploratory analysis, and are still far from providing a stand-alone method for reconstructing an accurate causal diagram. Moreover, our results support that the current best approach relies on an iterative *human-in-the-loop* process, based on the combination of CD methods and expert knowledge.

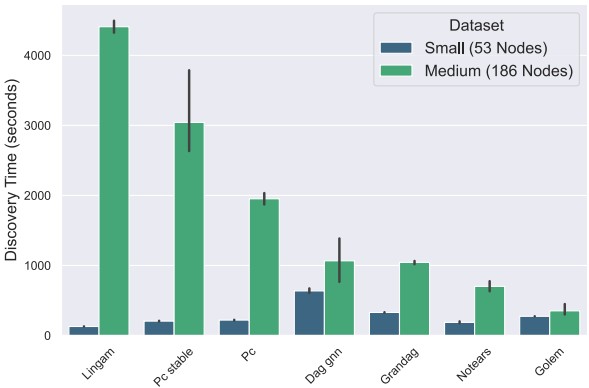

Figure 3: Time to discover a Causal graph with $n = 10.000$ samples. Methods thriving on CausalMan Small may be computationally impractical on CausalMan Medium.

# 7 CONCLUSIONS

Although much progress has been made in causal modeling, we have shown a number of limitations of methods for causal inference and discovery. We did so by introducing two novel causal benchmarks from manufacturing. The data is generated by a simulator based on physical models derived from first principles, and the integration of domain knowledge from experts received the highest priority when building the DGP. We envision that our benchmarks will serve as a playground to build causal models that can tackle the complexity of the real world, where most assumptions made by causal models no longer hold.

**Limitations:** From a simulation perspective, even though we modeled the system with a high degree of realism, it still inherits all the modeling assumptions of the underlying SCMs. From a benchmarking perspective, we did not test the most complex queries possible since, they are out of reach for all tested models. Although the chosen inference tasks are simple, the models performed far from optimal. These results indicate where their potential lies. On a conceptual level, the queries of interest during inference depend on the capabilities of the available models, and deeper analyses are possible as we develop new models capable of more advanced tractable inferences. Therefore, the advantage of having models that can represent complex interventional and counterfactual distributions, and not directly targeting ATE or CATE, lies in the inferences that become possible. These causal models, can open the door for enhanced explainability, counterfactual analyses (Janzing et al., 2020) and out-of-distribution generalisation (Richens & Everitt, 2024). Furthermore, accurate estimates of ATE or CATE may not always be enough to satisfy real-world use cases. Finally, learning-based causal models have a stronger scaling potential to high-dimensional settings where causality is applied, for example, to vision or multimodal data (Li et al., 2023).

**Is prior knowledge necessary?** Many of our experiments involved either estimating treatment effects or discovering causal relations that are trivial to domain experts. Yet, all tested models are far from providing an accurate answer both for Causal Inference and Discovery. Methods relying on *neuro-symbolic AI* could provide a way (Ahmed et al., 2022) to inject this knowledge on the model.

**Future Work:** We plan to extend our work by conducting deeper analysis in different directions. From a model-related perspective, it could be expanded by including non-parametric models (Friedman & Nachman, 2000; Cevid et al., 2020), tractable circuits (Zecevic et al., 2021; Poonia et al., 2024), and tailored CATE estimators (Athey et al., 2019). On the bench-marking side, further insights can be gained by performing new case studies focused on counterfactual quantities, and on multiple and/or unknown interventions (Jaber et al., 2020), and root-cause analysis. Further, we also aim to scale our simulator to even larger Causal Graphs.

## 8 REPRODUCIBILITY STATEMENT

We followed different procedures for ensuring the reproducibility of our experiments.

- We will release the simulator code, to enable researchers to generate new observational and interventional data (hard/soft-interventions, multiple interventions, and interventions on hidden variables).
- We release every dataset used for our experiments, including their complete causal graph and the marginalised ADMG.
- In Sec. F we describe the complete procedure used to embed numerically the data and to normalise it.
- Additional pre-processing of the data is made for some specific models in order to adapt them to work with our data is contained in Sec. F.
- The Hardware used to run every experiment is listed in F.
- Specific hyperparameters and modifications applied to models are completely listed in F.4, including where the code for each model has been taken.

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

## A    DATASETS RELEASE

All the data used in this paper, and more, is available at this link: Link to Zenodo anonymous repository.

## B    TASK DESCRIPTION

In this section we specify the tasks with an higher level of detail. In both tasks, the outcome variable $Y = Sec\_C2\_Machine1\_ProcessResult$. When conditioning, the evidence variable is called $HU\_HU\_Block\_Type\_ID\_num$, which will be assumed to be observed with value 921.

**Task 1:**

$$\text{ATE} = \mathbb{E}[Y|do(PF\_M1\_T1\_Force\_MpGood = 0)] - \mathbb{E}[Y|do(PF\_M1\_T1\_Force\_MpGood = 1)] \tag{4}$$

**Task 2:**

$$\text{ATE} = \mathbb{E}[Y|do(PF\_M1\_T1\_Force\_LTL = 18000)] - \mathbb{E}[Y|do(PF\_M1\_T1\_Force\_LTL = 15000)] \tag{5}$$

**Task 3:**

$$\text{ATE} = \mathbb{E}[Y|do(PF\_M1\_T1\_Force = 30000)] - \mathbb{E}[Y|do(PF\_M1\_T1\_Force = 16000)] \tag{6}$$

In this third task, the treatment increases the Force value to an extreme level with respect to nominal values for some product types, and the control intervention is instead in the desired range. The force variable has multiple bi-directed edges with other variables describing the PF process. Moreover, it is also a direct parent of other variables, therefore an extreme intervention can cause extreme values to propagate on other physical quantities that depend on it (For example $s_{grad}$ and $F_{max}$). After conditioning, we are now intervening for a HU type where the Upper Tolerance Limit is 28.000, therefore all product should now end up being scrapped.

## C    CAUSAL MECHANISMS

We proceed by describing the details of the DGP.

### C.1    CONDITIONAL DEPENDENCIES:

Given a node $n_1$, its distribution may depend on the value of one or more categoricals such as the supplier or the component type. For a node $n_1$ depending on a single categorical A, we can write it mathematically as

$$n_1 \sim \begin{cases} \mathcal{N}(\mu_0, \sigma_0) & \text{if } A = a_0, \\ \mathcal{N}(\mu_1, \sigma_1) & \text{if } A = a_1, \end{cases} \tag{7}$$

where $\mu_i$ and $\sigma_i$ are the mean and standard deviation of two Gaussian distributions, with $\mu_0 \neq \mu_1$ and $\sigma_0 \neq \sigma_1$. In Fig. 4, we provide a graphical illustration for a simple conditional dependency.

### C.2    STRUCTURAL EQUATIONS

Hereby we provide a more in-depth description of the production process, along with its relative physical description and structural equations. For more in-depth mathematical derivations, we address the interested reader to Budynas & Nisbett (2008) and Eslami et al. (2013).

$$A = a_1 \implies B|A = a_1 \sim \mathcal{N}(\mu_1, \sigma_1^2)$$

$$A = a_2 \implies B|A = a_2 \sim \mathcal{N}(\mu_2, \sigma_2^2)$$

$$A = a_3 \implies B|A = a_3 \sim \mathcal{N}(\mu_3, \sigma_3^2)$$

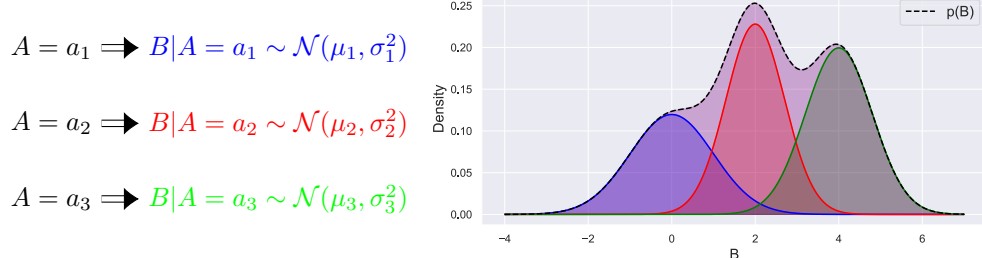

Figure 4: Example of a conditional dependency where A (categorical) determines the distribution of B. Node distributions are often not fixed a-priori, and their parameters are determined by the value of a number of categorical (parent) variables. The resulting marginal distribution can be asymmetric and multimodal.

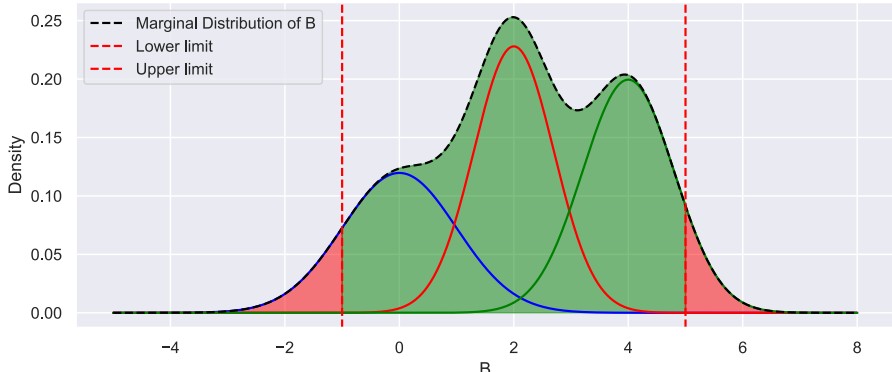

Figure 5: Given a *monitored* variable B, a monitoring mechanism checks if its value lies within an ideal range defined by the interval [B_LTL, B_UTL]. If yes, a binary r.v. B_MpGood will be *True*, signaling that the attribute is conformal, otherwise *False*. At the end of production, all the MpGood variables are aggregated into a ProcessResult variable via a logic AND operation, which consequently signal if the whole production process did run successfully.

**Model of a Magnetic Valve:**   A magnetic Valve is modeled by different parameters that describe its geometric and material properties. The Parameters are $E_{mv}$, describing the material elasticity of the valve, $A_{leak_{MV_{raw}}}$ describing the leakage area before starting production (A supplier may give us faulty MVs), $D_{mvMax}$ describing the maximum diameter, and $D_{mvMin}$ describing the minimum diameter, and $L_{mvPF}$ describing the axial length of the MV, coinciding with the optimal engagement length between the MV and the bore during the PF process.

All those parameters are not fixed, and are indeed randomly sampled from a distribution which conditionally depend on the type and supplier of the MV. Each combination of supplier and MV type implies a different node distribution for those parameters. This mechanism is a conditional dependency as described in C. Those conditional dependencies cause the marginal node distribution of those parameters to be multimodal and asymmetric. In other words, conditional dependencies induce a mixture model on the marginal node distributions.

**Model of an Hydraulic Unit:**   An HU is modeled with the same approach as for a MV. Indeed, an HU has the parameters $E_{hu}$ describing its elasticity and a $Force_{Lim}$ describing the force which is necessary to cause a non-zero leakage area.

On each HU we have different *chambers*, and every chamber has a certain number of *bores*. We model *each individual* bore in the HU with a set of parameters. Specifically, we have $E_{bore}$ describing the elasticity of the bore, $D_{boreMax}$ and $D_{boreMin}$ describing its maximum and minimum diameter. In this case, conditional dependencies appear both for the general HU parameters $E_{hu}$ and $Force_{Lim}$, but also in the parametrization of each individual bore.

**Intrinsic Magnetic Valve Leakage:**   A magnetic valve could be manufactured in a faulty way, resulting in *intrinsic leakage* through the valve, even in the "closed state". If quality control of the MV supplier works well, this intrinsic leakage should be zero. However, it may also happen that a magnetic valve gets damaged during assembly (e.g. due to high forces during press-fitting), leading to leakage through the valve itself. The initial intrinsic leakage of the valve as delivered by the supplier is modeled using $A_{leak_{MV}}$. As small intrinsic leakages are more likely than high values, and as the leakage area is continuous, we modeled a probability distribution for $A_{leak_{MV_{raw}}}$ and then used a ReLU function to cut off unrealistic negative leakage area values.

$$A_{leakMV} = \text{ReLU}(A_{leak_{MV_{raw}}}) \tag{8}$$

**Total Leakage Area of a Chamber**   The total leakage area of a chamber in the Hydraulic Unit block is the sum of the leakage areas of each bore/Magnetic Valve in the chamber

$$A_{leak_{tot}} = A_{leak_{Bore_1}} + A_{leak_{Bore_2}} + \ldots, \tag{9}$$

where $A_{leak_{Bore}}$ is the total leakage are per bore/Magnetic valve.

The fluid is assumed to be able to take two different leakage paths, one through the valve itself ($A_{leak_{MV}}$, see below for details) and one through the Press-Fitting connection ($A_{leak_{PF}}$). Therefore, for a single bore, the total leakage area is the sum of the leakage though the MV and through the PF.

$$A_{leak_{Bore}} = A_{leak_{MV}} + A_{leak_{PF}} \tag{10}$$

**Leakage area and geometry of the Press-Fitting Connection**   The leakage area through the Press Fitting connection $A_{leak_{PF}}$ depends mainly on the geometry of the bore and Magnetic Valve. As the cylindrical surfaces are not perfectly round, we assume an interval for the maximum ($D_{mvMax}, D_{boreMax}$) and minimum diameter ($D_{mvMin}, D_{boreMin}$), respectively. When studying the unwanted leakage of fluid, it is important to consider the difference between minimum and maximum diameters, identified as $\Delta D$, as the may have negative consequences for the press-fitting process and result in the scrap of a product.

$$\Delta D_{\text{max}} = D_{\text{mvMax}} - D_{\text{boreMin}},$$
$$\Delta D_{\text{min}} = D_{\text{mvMin}} - D_{\text{boreMax}}. \tag{11}$$

To account for effects from the machine on the resulting leakage (such as acentric positioning of the valve with respect to the bore during press fitting), we introduce a machine dependent limit for resulting leakage ($LeakTolMachine$). When $\Delta D$ is higher than the threshold $LeakTolMachine$,

we observe a leakage (area) through the press-fitting. This phenomenon can be modeled also with a ReLU function as follows

$$\begin{aligned} \Delta D_{Leak_{min}} &= \Delta D_{min} - \text{LeakTolMachine} \\ \Delta D_{Leak_{max}} &= \Delta D_{max} - \text{LeakTolMachine} \end{aligned} \implies \begin{aligned} A_{Leak_{min}} &= \text{ReLU}(\Delta D_{Leak_{min}}) \\ A_{Leak_{max}} &= \text{ReLU}(\Delta D_{Leak_{max}}) \end{aligned} \tag{12}$$

Moreover, in real production lines, it is likely that different press-fitting machines have a different threshold for leakage due to badly adjusted press fitting processes. Additionally, using the coefficient $\beta_{asym}$ we can model how much the total leakage area is affected by $\Delta D_{Leak_{Min}}$ and $\Delta D_{leak_{Max}}$, respectively.

$$A_{\text{leakPF}} = \beta_{\text{asym}} A_{Leak_{max}} + (1 - \beta_{\text{asym}}) A_{Leak_{min}}, \tag{13}$$

where $\beta_{asym} = 1$ means that only the maximum leakage Area $A_{leak_{MV}}$ is effective, a value of 0.5 means that minimum and maximum leakage area are weighted equally.

**The Press Fit process** The PF machine applies a force which inserts the MV into a bore of the HU. Apart from inserting the MV into the bore, the force will also deform the bore. At the end of the process the bore will be deeper than before by a certain amount which is determined by the physical models (with some stochasticity). Part of the deformation is permanent, and another other part will disappear after the pressing force is removed at the end of the process, as it is related to the elasticity of the material. If the force is too high, we may cause a damage that will end in the component being scrapped. We start by defining the effective elasticity modulus $E_{\text{eff}}$ as

$$E_{\text{eff}} = \left( \frac{1}{E_{\text{bore}}} + \frac{1}{E_{\text{mv}}} \right)^{-1} \tag{14}$$

where $E_{\text{bore}}$ is the elasticity of the bore and $E_{\text{mv}}$ the elasticity of the MV. The effective elasticity is used to define the stiffness of the press-fitting machine as

$$K_{\text{stiffPF}} = K_{\text{stiffPF}_{Ref}} \cdot \frac{\Delta D_{\text{mean}}}{K_{\text{stiffPF}_{\Delta D_{Ref}}}} \cdot \frac{E_{\text{eff}}}{K_{\text{stiffPF}_{E_{Ref}}}}, \tag{15}$$

where $K_{\text{stiffPF}_{Ref}}$, $K_{\text{stiffPF}_{\Delta D_{Ref}}}$, and $K_{\text{stiffPF}_{E_{Ref}}}$ are new machine-dependent parameters describing how much the reference stiffness of the PF machine $K_{\text{stiffPF}_{Ref}}$ varies linearly with $\Delta D_{\text{mean}}$ and $E_{\text{eff}}$. As before, those reference parameter are not absolute and may vary across different PF machines. Moreover, in 15 $\Delta D_{\text{mean}}$ is modeled similarly to Eq.13, where we use $\beta_{asym}$ again to balance how much the PF process is affected by the maximum and minimum diameter,

$$\Delta D_{\text{mean}} = \beta_{\text{asym}} \Delta D_{\text{max}} + (1 - \beta_{\text{asym}}) \Delta D_{\text{min}}. \tag{16}$$

Now we have all the quantities which are necessary to compute the total stiffness $K_{\text{stiff}}$ of the system,

$$K_{\text{stiff}} = \left( \frac{1}{K_{\text{stiffMachine}}} + \frac{1}{K_{\text{stiffPF}}} \right)^{-1} \tag{17}$$

where $K_{\text{stiffMachine}}$ is the stiffness deriving from the machine itself, and $K_{\text{stiffPF}}$ is the stiffness coming from the press-fitting operation. Using $K_{\text{stiffPF}}$ it is possible to derive the pressing force as

$$\text{Force} = L_{\text{mvPF}} \cdot K_{\text{stiffPF}} \tag{18}$$

where we used the axial length of the MV $L_{\text{mvPF}}$, as it coincides with how much the MV should be inserted into the HU with PF. By dividing the Force by the stiffness of the system $K_{\text{Stiff}}$, we can compute the difference in vertical position of the PF tool before and after the operation, which coincides with the permanent deformation (in depth) of the component,

$$\Delta s_{\text{grad}} = \frac{\text{Force}}{K_{\text{stiff}}}. \tag{19}$$

We remark that $\Delta s_{\text{grad}}$ also coincides wit the difference in position of the tool before and after the maximum pressing force is achieved and removed. Therefore, it does not include any elastic effect of the material, which may be present only while the pressing force is still present. The quantity above can be used to compute the final position of the tool $s_{\text{grad}}$

$$s_{\text{grad}} = s_0 + \Delta s_{\text{grad}} \tag{20}$$

where $s_0$ is, instead, the position of the PF tool at the beginning of the process.

**Maximum forces and dispacement on a single bore:**   As written above, during PF multiple forces are applied to insert all MVs into the HU. Focusing on the maximum force $F_{\max}$ achieved on a single bore/MV pair, we can decompose it on the optimal $Force$ variable, plus another variable $\Delta F_{\text{trigger}_{\text{stop}}}$ describing how much the force went over the value Force, before a trigger in the machine did stop the operation.

$$F_{\max} = \text{Force} + \Delta F_{\text{trigger}_{\text{stop}}}, \tag{21}$$

where $\Delta F_{\text{trigger}_{\text{stop}}}$ is randomly sampled. The reason why we model the maximum force is because, if the applied force is too high, the component will be damaged and result in a leakage. Moreover, from the maximum bore force we can compute the maximum difference in displacement of the tool during the PF process, written as

$$\Delta s_{\max} = \frac{\Delta F_{\text{trigger}_{\text{stop}}}}{K_{\text{stiffMachine}}}, \tag{22}$$

which, with respect to $\Delta s_{\text{grad}}$, includes also the elastic deformation which will disappear after the force is removed. Thanks to $\Delta s_{\max}$ we can get the absolute maximum displacement of the tool,

$$s_{\max} = s_{\text{grad}} + \Delta s_{\max}. \tag{23}$$

The maximum displacement $s_{\max}$ during the process includes both the actual deformation of the component, but also an elastic deformation which will disappear once the pressing force is removed.

**Maximum Forces and Displacement:**   Forces applied during PF cannot be higher than a machine and product-dependent threshold $F_{\text{lim}}$, otherwise we might incur in a damage of the components. First, we define $F_{\max}$ as the highest value achieved among all maximum forces in the chamber's bores. Then, we can compute how much the maximum force went over the limit with

$$\Delta\text{Force} = F_{\max} - F_{\text{Lim}} \quad \implies \quad \Delta\text{Force}_{\text{ReLu}} = \text{ReLU}(\Delta\text{Force}) \tag{24}$$

where we applied a ReLU again to make it zero if the force was below the limit. In order to model the relation between the applied forces and potential faults inducing a nonzero leakage area, we model the LeakTolMachine parameter as follows:

$$\text{LeakTolMachine} = \text{LeakTolMachine}_0 + \frac{\text{LeakTolMachine}_{\text{REF}} \cdot \Delta\text{Force}_{\text{ReLu}}}{\Delta\text{Force}_{\text{REF}}} \tag{25}$$

where we made explicit the dependence on $\Delta\text{Force}_{\text{ReLu}}$. Lastly, we have similar machine parameters $\text{LeakTolMachine}_0$ to model the minimum tolerance, plus $\text{LeakTolMachine}_{\text{REF}}$ and $\Delta\text{Force}_{\text{REF}}$ to model the dependence on $\Delta\text{Force}_{\text{ReLu}}$.

# D   ADDITIONAL RESULTS

In this section we provide a more exhaustive exposition of our results for performance and runtime.

| Causal Model | ATE MSE | CATE MSE | JS-Div Tr. | MSE | MMD |
|---|---|---|---|---|---|
| CBN | 0.659 (0.001) | **0.036 (0.007)** | 0.136 (0.092) | 0.259 (0.186) | 0.702 (0.121) |
| NCM | 0.631 (0.015) | 0.049 (0.028) | 0.233 (0.040) | 0.307 (0.033) | **0.086 (0.000)** |
| CAREFL | 0.652 (0.014) | 0.175 (0.106) | 0.512 (0.093) | **0.086 (0.073)** | Nan |
| CNF | **0.631 (0.015)** | 0.065 (0.063) | 0.156 (0.047) | 0.299 (0.093) | Nan |
| VACA | 0.648 (0.015) | 0.230 (0.270) | **0.033 (0.009)** | 0.059 (0.017) | 0.128 (0.000) |
| Linear r. | 1e8 (1e10) | - | - | - | - |
| Logistic r. | 0.698 (0.066) | - | - | - | - |

Table 3: Comparison between models for the second treatment effect estimation task on CausalMan Small with $n = 50.000$ samples and ground truth ADMG. Instabilities during sampling prevented to evaluate MMD for CNF and CAREFL, as multiple datapoints diverged to $+\infty$ as a results of training instabilities.

| Causal Model | ATE MSE | CATE MSE | JS-Div Tr. | MSE | MMD |
|---|---|---|---|---|---|
| NCM | 1.115(0.118) | 1.665(0.159) | 0.206(0.005) | **0.172(0.001)** | 0.259(0.018) |
| CAREFL | **0.982(0.223)** | **1.539(0.635)** | 0.164(0.105) | 0.279(0.197) | Nan |
| CNF | 1.218(0.012) | 1.784(0.082) | 0.297(0.003) | 0.535(0.007) | Nan |
| VACA | 1.214(0.009) | 1.890(0.163) | **0.163(0.003)** | 0.265(0.006) | **0.244(0.009)** |
| Linear r. | 4.748(0.142) | - | - | - | - |
| Logistic r. | 0.992(0.015) | - | - | - | - |

Table 4: Comparison between models for the third treatment effect estimation task on CausalMan Small with $n = 50.000$ samples and ground truth ADMG. Linear regression in this case is clearly disadvantaged due to the presence of hidden confounders and nontrivial causal mechanisms.

| Causal Model | ATE MSE | CATE MSE | JS-Div Tr. | MSE | MMD |
|---|---|---|---|---|---|
| NCM | 0.580 (0.043) | 0.067 (0.052) | 0.179 (0.016) | 0.257 (0.017) | 0.380 (0.008) |
| CAREFL | **0.614 (0.009)** | **0.033 (0.015)** | **0.054 (0.038)** | **0.098 (0.069)** | **0.212 (0.023)** |
| CNF | 0.618 (0.006) | 0.062 (0.036) | 0.127 (0.056) | 0.218 (0.096) | 0.335 (nan) |
| Linear r. | 2e9 (2e9) | - | - | - | - |
| Logistic r. | 0.649 (0.119) | - | - | - | - |

Table 5: Comparison between models for the second treatment effect estimation task on CausalMan Medium with $n = 20.000$ samples and ground truth ADMG.

# E   EXPERIMENT SETTING

## E.1   METRICS

We can write the *Structural Hamming Distance* SHD between a graph $\mathcal{G}$ with adjacency matrix $A$ and the ground truth $\mathcal{G}^\star$ with adjacency matrix $A^\star$ as:

$$SHD(A, A^\star) = \sum_{i,j=0}^{n} \mathbf{I}_{A_{ij} \neq A_{ij}^\star} \tag{26}$$

Since discovering an individual edge can be thought as a binary classification task (edge/no-edge), it is common to measure metrics such as precision and recall:

$$Pr = \frac{tp}{tp + fp}, \qquad\qquad Rec = \frac{tp}{tp + fn}. \tag{27}$$

where *tp* stands for true positives, *fp* for false positives and *fn* for false negatives.

## E.2   CAUSAL MODELS

Here we provide a more detailed description of the tested causal models.

- **Causal Bayesian Networks:** For Bayesian Networks (BN), edges do not have a causal semantic, and they are indeed only an observational Layer 1 model. However, it is possible to define a do-operator for Bayesian Networks, and obtain an interventional L2 model called *Causal Bayesian Network* (CBN) (Bareinboim et al., 2022).
- **Neural Causal Models:** Presented by Xia et al. (2022a), *Neural Causal Models* (NCM) consist in a SCM where each structural equation is parameterized by a neural network. NCMs, as a special case of SCMs, are Layer 3 models capable of answering counterfactual queries, when identifiable (Xia et al., 2022b). More info about our implementation in F.4.
- **CAREFL:** *Causal AutoREgressive normalizing Flows* (CAREFL) (Khemakhem et al., 2021), uses Normalizing flows with affine layers and the Causal Ordering to answer queries up to the counterfactual level.

| Causal Model | ATE MSE | CATE MSE | JS-Div Tr. | MSE | MMD |
|---|---|---|---|---|---|
| NCM | 1.629 (0.031) | 1.271 (0.031) | 0.589 (0.000) | 1.000 (0.000) | 0.389 (0.007) |
| CAREFL | 1.730 (0.068) | **1.199 (0.149)** | **0.351 (0.028)** | **0.780 (0.034)** | 0.185 (0.022) |
| CNF | 1.822 (0.016) | 1.347 (0.052) | 0.357 (0.088) | 0.783 (0.099) | 0.212 (0.159) |
| Linear r. | **0.297 (0.019)** | - | - | - | - |
| Logistic r. | 1.362 (0.016) | - | - | - | - |

Table 6: Comparison between models for the first treatment effect estimation task on CausalMan Medium with $n = 20.000$ samples and ground truth ADMG.

| Method | SHD | Prec. | Rec. | SID | p-SD |
|---|---|---|---|---|---|
| PC | 144.2 (0.837) | **0.123 (0.014)** | 0.056 (0.007) | 2208.2(40.935) | 0.099(0.043) |
| PC-Stable | 127.4 (1.949) | 0.072 (0.052) | 0.017 (0.012) | **2118.4(78.904)** | 0.017(0.004) |
| DAG-GNN | 147.8 (13.479) | 0.008 (0.017) | 0.002 (0.004) | 2275.8(32.568) | 0.038(0.017) |
| NOTEARS | 137.8 (1.922) | 0.018 (0.028) | 0.005 (0.007) | 2280.4(14.398) | 0.078(0.015) |
| GOLEM | 263.2 (19.791) | 0.043 (0.015) | **0.063 (0.024)** | 2371.8(40.258) | 0.427(0.003) |
| LiNGAM | 212.2 (31.196) | 0.043 (0.014) | 0.043 (0.022) | 2271(34.655) | 0.278(0.028) |
| GranDAG | **116 (2.646)** | 0.022 (0.049) | 0.002 (0.004) | 2240.2(24.468) | **0.001(0.001)** |
| Random DAG | 208 (15.215) | 0.051 (0.017) | 0.050 (0.017) | 2260.8(75.652) | 0.413(0.026) |

Table 7: Comparison for Causal Discovery on CausalMan Small (20.000 Samples).

- **Causal Normalizing Flows:** (Javaloy et al., 2023) provided a generalisation of CAREFL that uses the whole causal graph, includes non-additive noise models, and provides stronger identification guarantees, yielding Causal Normalizing Flows (CNF).
- **VACA:** Based on *Variational Graph Autoencoders* (Kipf & Welling, 2016), *Variational Causal Graph Autoencoder* (VACA) (Sanchez-Martin et al., 2021) provides a counterfactual model based on *Graph Neural Networks*.

# F IMPLEMENTATION DETAILS

In this supplementary section, we provide additional details on the architectures and implementations that have been tested. Furthermore, we list all the necessary modification that have been necessary to run the models with our datasets with hybrid data-types.

## F.1 DETERMINISM

Every experiment was run 5 different times with the random seeds 4, 6, 42, 66 and 90.

## F.2 HARDWARE

To perform a fair experimental evaluation of their tractability, each run was performed on a A100 GPU with 80 GB of GPU memory allocated, and one core of a ADD CPU, with approximately 300000 GB of RAM memory allocated.

Not all methods can leverage GPU parallelisation, therefore:

- For Causal Inference, regression-based techniques and CBNs are run using only CPUs.
- For Causal Discovery, PC algorithm, PC-Stable, NOTEARS, and LiNGAM are run using only CPUs.

## F.3 DATA PREPROCESSING:

For running the chosen models, data had to be embedded in a numerical format. Therefore, categoricals and discrete variables have been converted to an ordinal encoding (1, 2, 3, etc.). After

| Method | SHD | Prec. | Rec. | p-SD |
|---|---|---|---|---|
| PC | 702.0 (3.24) | 0.015 (0.003) | 0.004 (0.001) | 0.061(0.05) |
| PC-Stable | 591.2 (0.83) | 0.020 (0.007) | 0.002 (0.001) | 0.002(0.001) |
| DAG-GNN | 580.8 (22.28) | 0.003 (0.006) | 0.000 (0.001) | 0.001(0.001) |
| NOTEARS | 580.2 (1.78) | 0.024 (0.026) | 0.002 (0.002) | 0.004(0.001) |
| GOLEM | 845.0 (113.00) | **0.028 (0.005)** | 0.012 (0.004) | 0.283(0.131) |
| LiNGAM | 960.2 (100.18) | 0.027 (0.015) | 0.016 (0.007) | 0.287(0.015) |
| GranDAG | **543.4 (2.88)** | 0.017 (0.037) | 0.000 (0.001) | **2.32e-5(3.79e-5)** |
| Random DAG | 1189.6 (9.83) | 0.020 (0.002) | **0.019 (0.002)** | 0.474(0.004) |

Table 8: Comparison for Causal Discovery on CausalMan Medium (20.000 Samples).

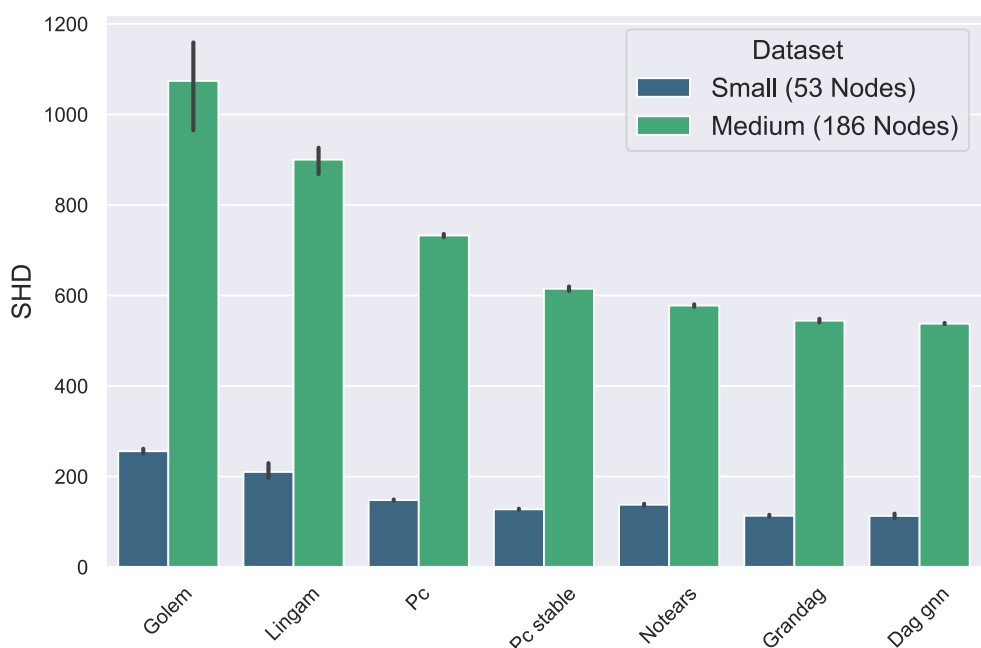

Figure 6: Difference in SHD between CausalMan Small and Medium.

obtaining a purely numerical dataset, every individual variable has been normalized via min-max normalization to be within the -1 and 1 range.

Models like CBNs are designed to work exclusively on discrete domain and are not tailored for hybrid datatypes. To overcome this limitation, CBNs have been fitted on a different version of the datasets where the continuous variables have been uniformly quantized.

For CNFs, CAREFL and VACA, data sampled from those models had to receive a binarization of the outcome variable and a binning of the conditioning variable. The binarization of the target variables has been done such that the target variable would be -1 if output was less than 0, and 1 if output higher than 0. For the conditioning variable, instead, the bins corresponded to the values of the evidence variable that were present in the training data, and the operation was necessary since the variable is discrete, otherwise it would have been impossible to evaluate empirically the conditional interventional distribution.

Among our tested models, Only NCMs are models that can adapt by design to hybrid datatypes, therefore they are the only ones that didn't necessitate any pre-processing for the training data apart from embedding of categoricals and data normalization. During estimation, interventional distributions were computed directly from the raw data that has been sampled from the estimated interventional distributions, without any post-processing.

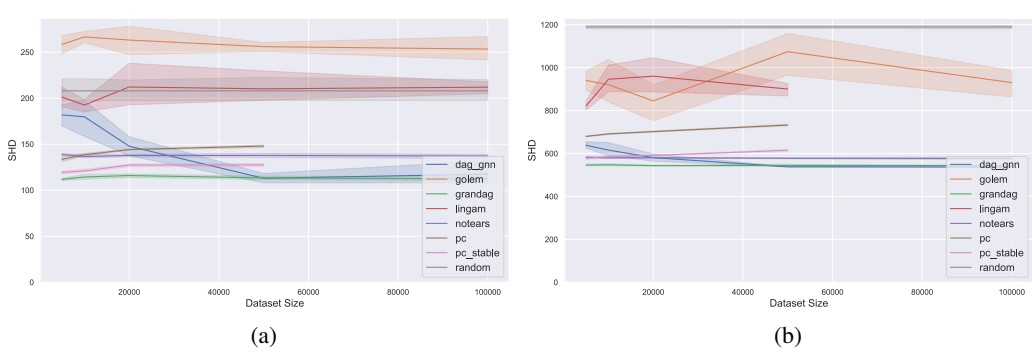

(a)                                    (b)

Figure 7: SHD as a function of dataset size for CausalMan Small (7a) and Medium (7b). Using more data has a minimal impact and is mostly detrimental to the overall Structural Hamming Distance.

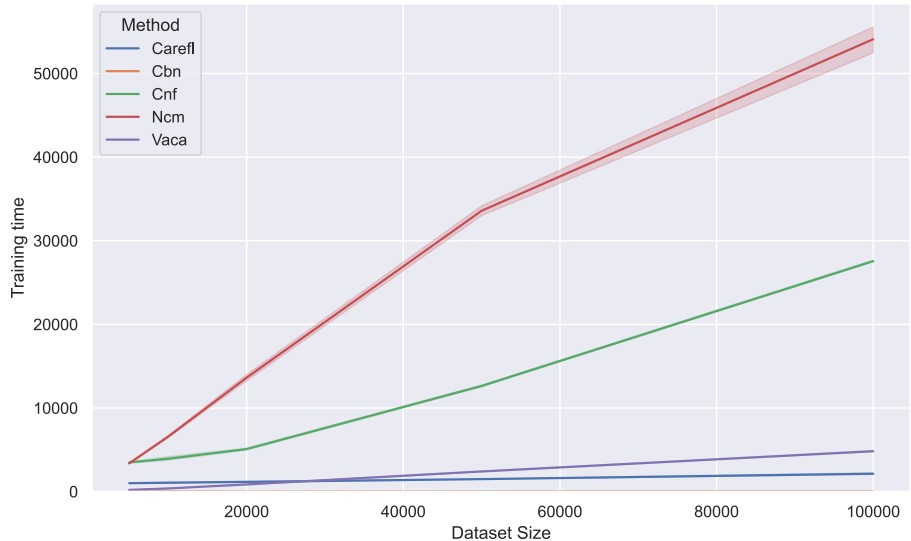

Figure 8: Average runtime (seconds) vs. dataset size for CausalMan Small.

### F.4    IMPLEMENTATION OF CAUSAL MODELS

For convenience, all the tested model have been incorporated into a configurable framework, present in this paper's supplementary material.

**Linear and Logistic Regression:**    For linear and logistic regression estimates, we used the implementations provided in the *DoWhy* python library.

**Causal Bayesian Networks (CNB):**    For CBNs, we use the implementation contained in the *pgmpy* python library. The score function that has been used it the K2.

**Neural Causal Models (NCM):**    We used the original implementation contained in Github Link, and applied minor modifications in order to adapt the model to handle hybrid data-types. Modifications have been made because, for each individual parameterized structural equation, NCMs require architectures capable of estimating conditional distributions $p(v_i|Pa_{\mathcal{G}}(v_i), u_i)$, as their log-likelihood is used for training Xia et al. (2022a). In detail, binary variables have been modeled using

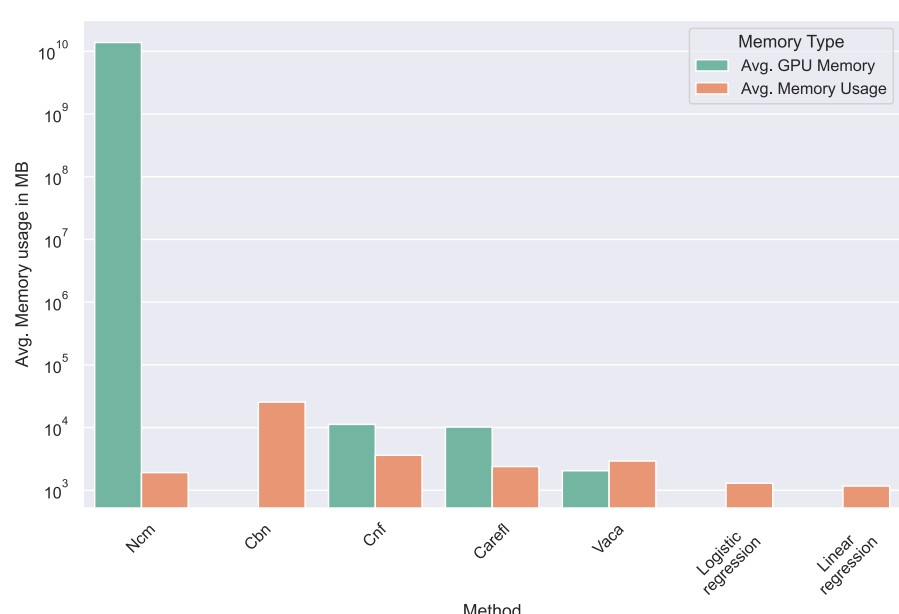

Figure 9: Bar plot showing the memory usage (RAM and GPU) for CausalMan Medium.

MADE, as in the original paper. The MADE implementation we use is taken from: Github Link. For discrete/categorical variables, MADE is still used upon minor modifications to the architecture in order to adapt it to discrete and non-binary domains. Indeed, discrete variables have been one-hot-encoded, then fed to the neural network, which would output the logit values for each discrete value. The input size of MADE in this case would be, for a causal graph $\mathcal{G}$,

$$D = |Pa_{\mathcal{G}}(\mathbf{x}_i)| + |u_i| + |v_i|. \tag{28}$$

where the last $|v_i|$ variables consist in the one-hot-encoding of the realisation of $v_i$.

Finally, structural Equations for Continuous variables are parameterised using Conditional Normalizing Flows Winkler et al. (2023)

**Causal Normalizing Flows, CAREFL & VACA:** The implementation that has been used is Link to GitHub Repository.

### F.5 HYPER-PARAMETERS AND TRAINING SETTINGS

To ensure reproducibility of every experiment, we list here all the modification applied to every single causal model and causal discovery method.

#### F.5.1 SETTINGS FOR CAUSAL MODELS

We reflect the implementation used in the original papers for all Causal Models tested. However, given the large size of the dataset in terms of covariates and number of datapoints, we apply the following modifications, mostly to increase the number of parameters and capacity for each model. Modifications are as follows:

- **CAREFL and Causal Normalizing Flows**: For both models, we did increase their size to have 4 layers with 64 hidden nodes each. Training optimization parameters are not changed with respect to the paper (Javaloy et al., 2023).
- **VACA**: 300 training epochs and batch-size of 1024. Both encoder and decoder use the *Graph Isomorphism Network* (GIN) (Xu et al., 2019) version of VACA. The encoder uses 2

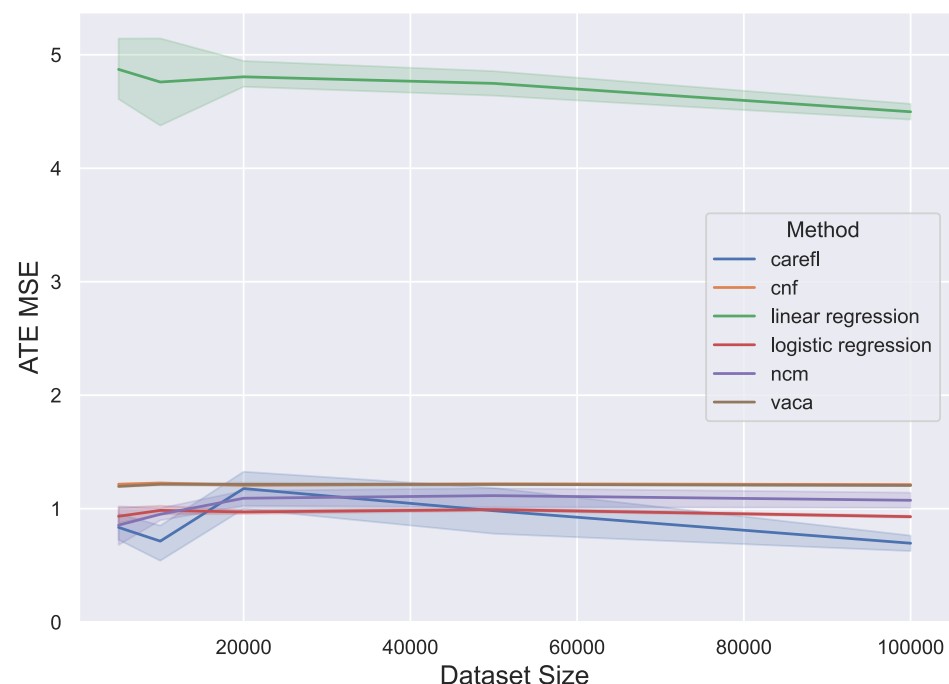

Figure 10: Line plot showing ATE MSE performance vs. Dataset size on the third ATE task. On nontrivial tasks with large amount of nonlinearities and confounders, linear regression is clearly in disadvantage.

hidden layers. The inner dimensionality is always 64. Even tough design conditions require to have a number of layers proportional to the diameter of the graph, scaling attempts to make the model bigger resulted in loss of convergence during training, and are a common limitation of Graph Neural Networks.

- **NCM**: For CausalMan Small, we use a batch-size of 1024 and 1.000 training epochs. For CausalMan Medium, we use a batch-size of 2048 and 600 training epochs. Training algorithm is still *AdamW* (Loshchilov & Hutter, 2019) with learning rate 0.004 and the *Cosine Annealing* scheduler with warm restarts.

### F.6 SETTINGS FOR CAUSAL DISCOVERY

All the tested models used the implementations present in the gcastle python library. All used Causal Discovery models reflect their original papers cited in 5.2 apart from the design choices listed below:

- **PC and PC-Stable:** The $^2$ Conditional Independence test was used.
- **NOTEARS:** The $L_2$ loss function was used.
- **GranDAG:** We used a batch-size of 1024 samples and 4 hidden layers, each one with 64 hidden nodes.
- **DAG-GNN:** We used a batch-size of 1024.

## G GROUND TRUTH CAUSAL GRAPHS

In this section we provide a visual depiction of all the ground truth causal graphs, both the complete graphs involved in the DGP and the partially observable ones obtained after a latent projection.

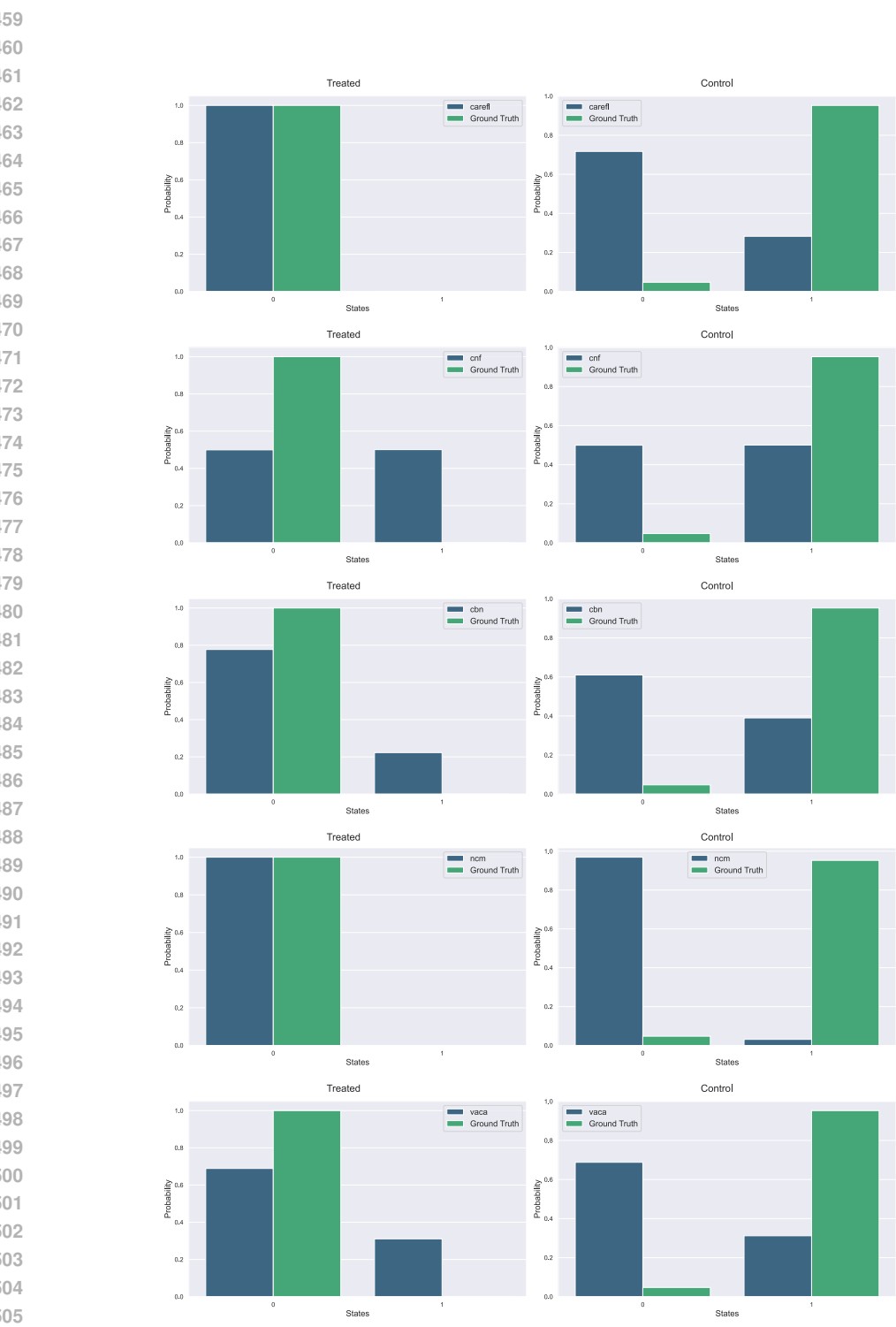

Figure 11: Estimated Interventional distributions for first ATE task on CausalMan Small (20.000 samples, seed 42). Causal models are not consistent when estimating interventional distributions, and cannot provide accurate reconstructions of both treated and control populations at the same time.

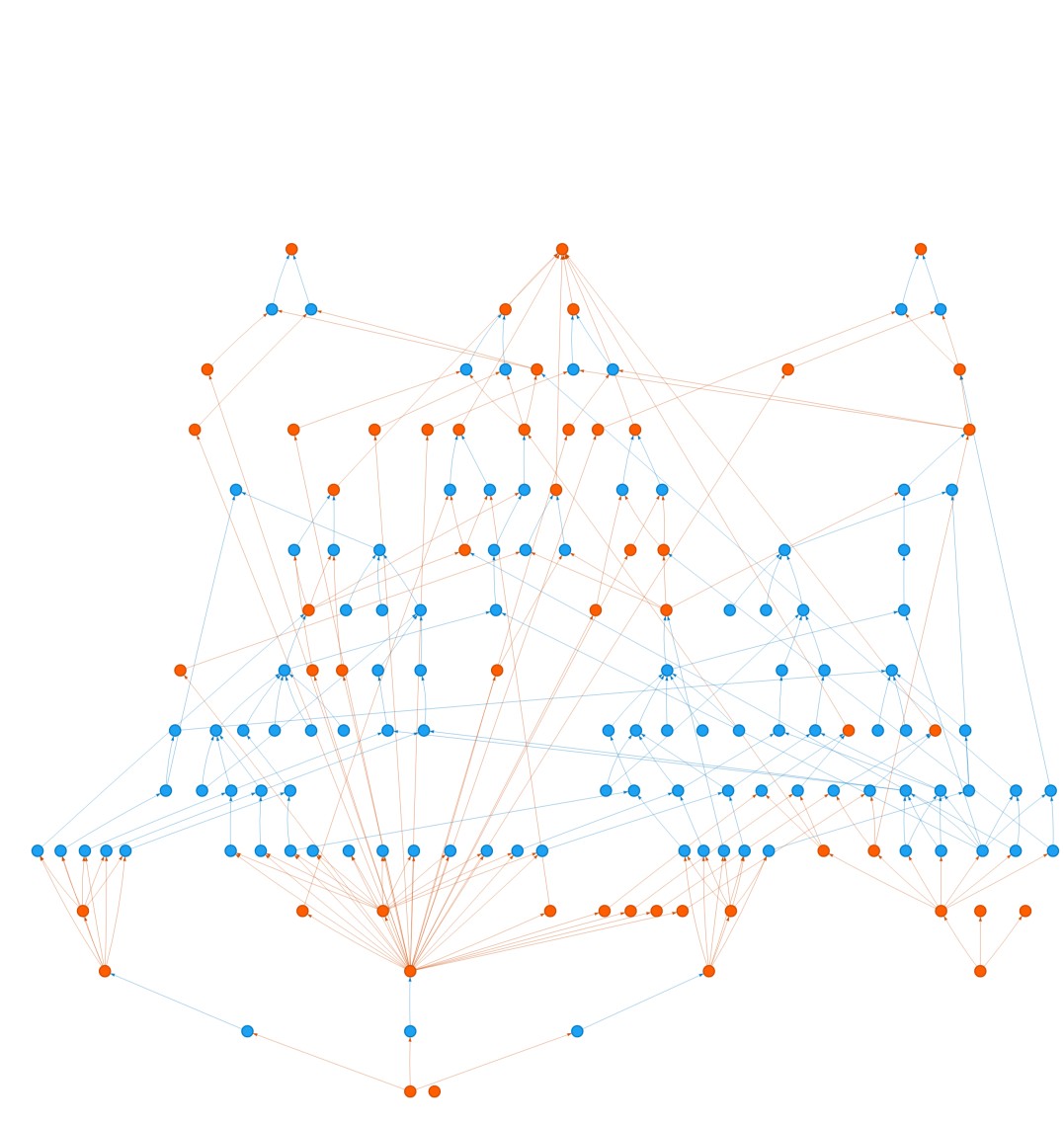

Figure 12: Complete Ground truth causal graph including hidden variables for CausalMan Small. Observable variables are colored in orange, and latent ones are colored in blue. Approximately 50-60 % of variables are latent.

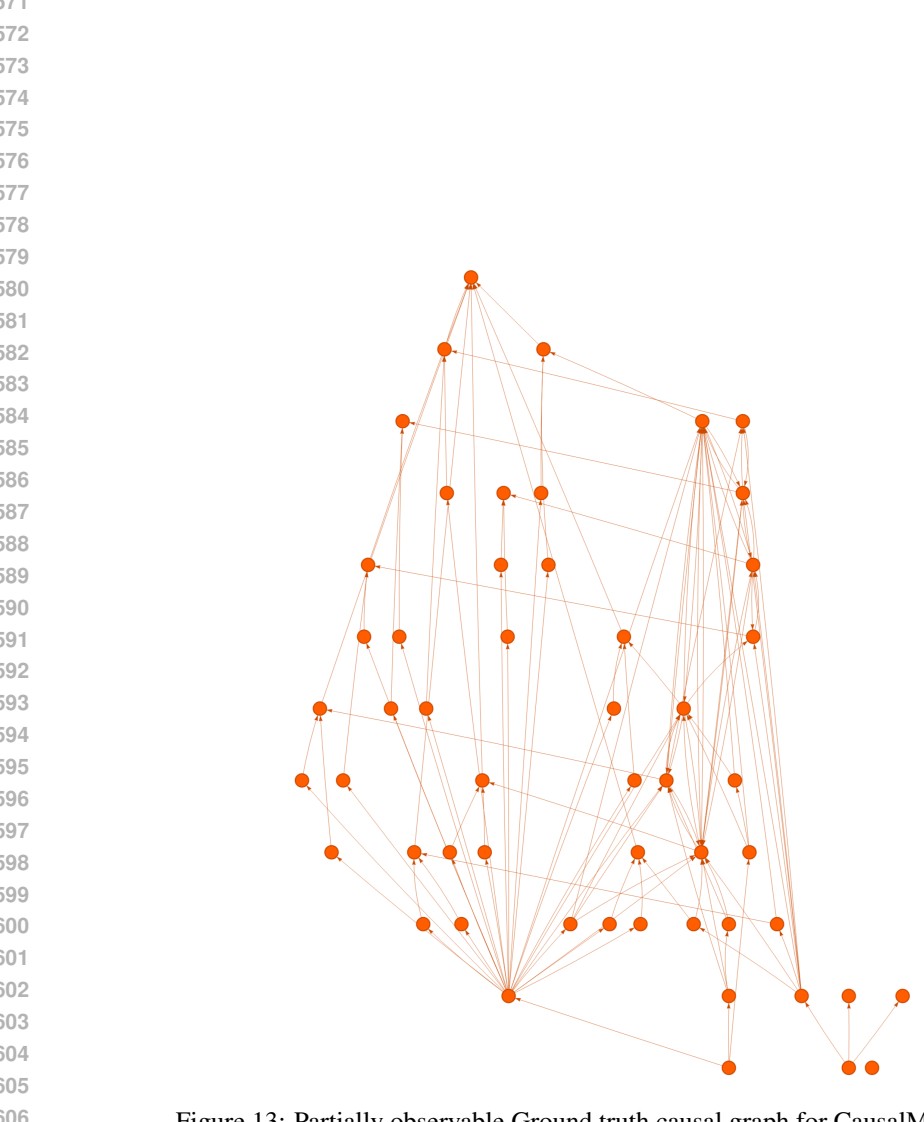

Figure 13: Partially observable Ground truth causal graph for CausalMan Small.

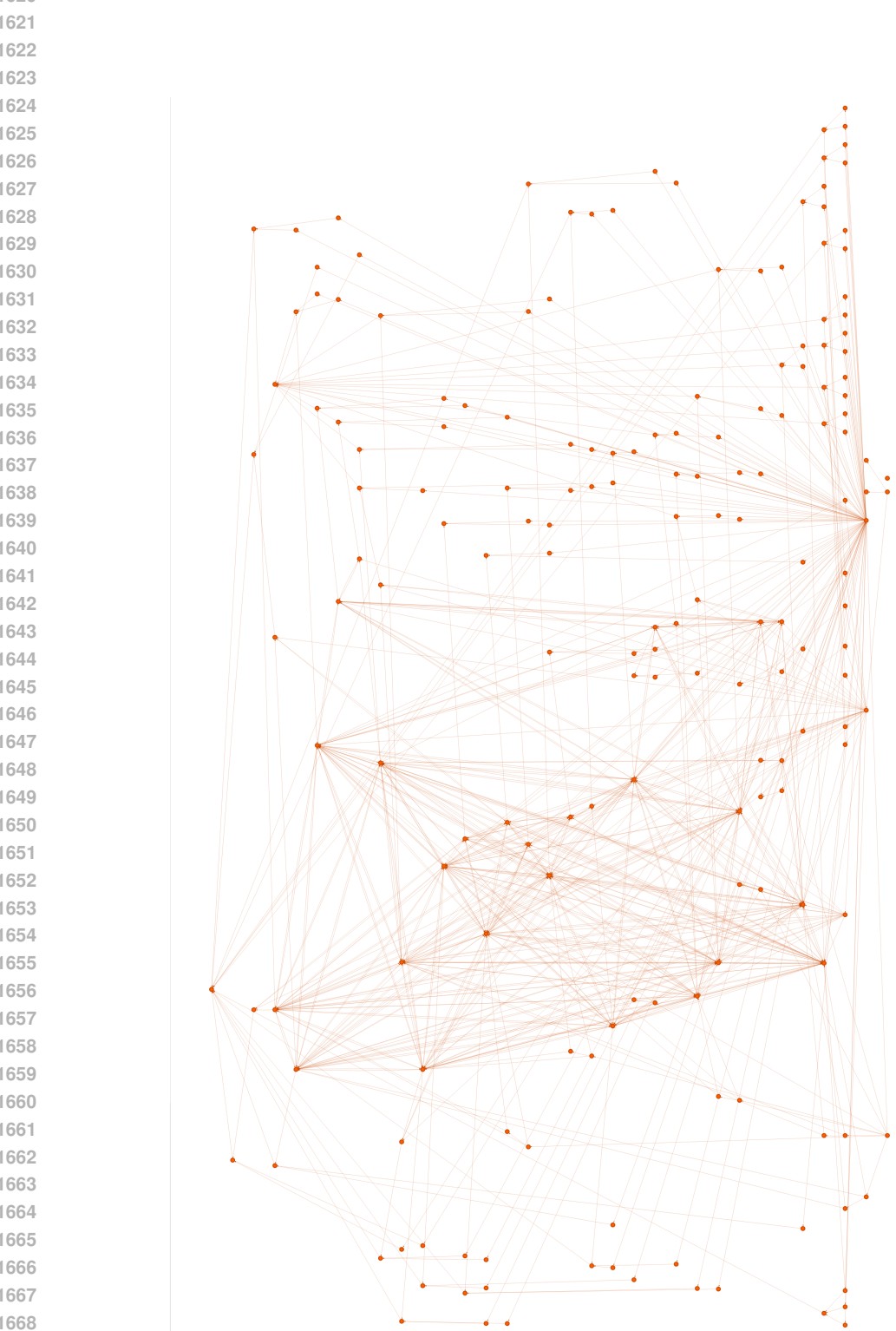

Figure 14: Partially observable Ground truth causal graph for CausalMan Medium.

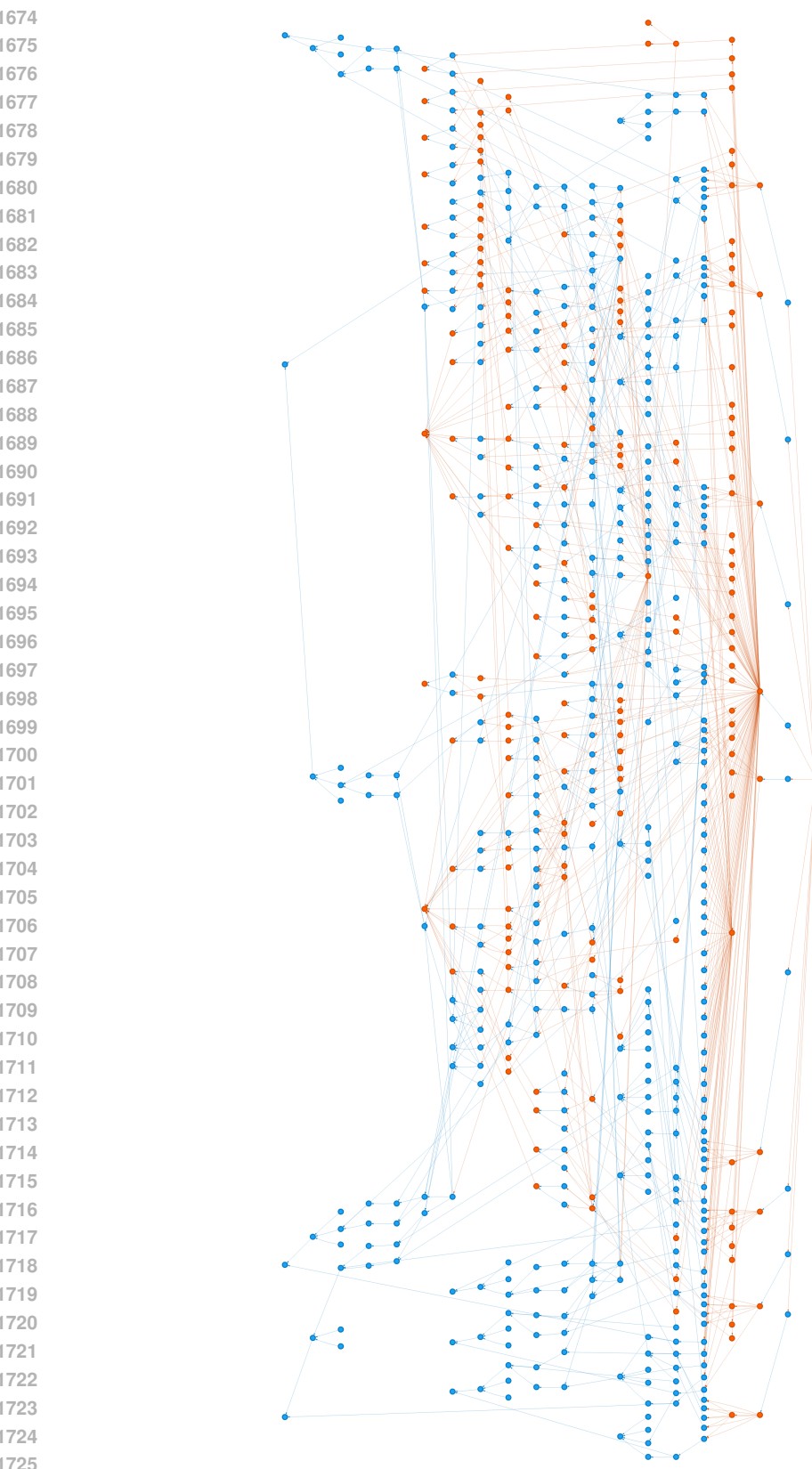

Figure 15: Complete Ground truth causal graph including hidden variables for CausalMan Medium. Observable variables are colored in orange, and latent ones are colored in blue. Approximately 50-60 % of variables are latent.

