# OpenReview forum: "The Challenging Growth: Evaluating the Scalability of Causal Models"
_ICLR.cc/2025/Conference — Submitted to ICLR 2025_

### Official Review · Reviewer_PVqx · 2024-10-30

**Soundness:** 2
**Presentation:** 2
**Contribution:** 3
**Rating:** 8
**Confidence:** 3

**Summary:**

The paper provides two datasets from simulators of a real-world manufacturing process. The simulators are large SCMs constructed with the support of domain experts, with different numbers of nodes (referred to as small and medium).

With the datasets, the authors provide a synthetic benchmark for different causal inference tasks, which address some shortcomings in other synthetic datasets, by providing (1) high-dimensional data (53 and 186 nodes), (2) causal insufficiency (both datasets have a large number of unobserved confounders), and (3) structural assignments, such as non-linearities, categorial variables and other artifacts, and sampling procedures (batching) which are not commonly studied in causal inference but may be realistic in manufacturing scenarios.

The authors then evaluate a cohort of causal models and causal discovery algorithms on the datasets.

**Strengths:**

The scarcity of meaningful benchmarks in causal inference is a serious problem. Efforts to produce new benchmarks from both real-world data or complex, well-motivated simulators are direly needed.

The authors make a valuable contribution to these efforts by providing datasets from what appears to be a complex, well-motivated simulator built together with domain experts, in an understudied field (manufacturing) that holds promise for the applications of causal inference, e.g., root cause analysis of manufacturing failures, etc. I appreciate the immense effort in building such a large simulator while integrating the feedback of domain experts.

In my opinion, the introduction and background are well-written and provide enough pointers to the relevant literature.

**Weaknesses:**

In my opinion, there are some weaknesses that depreciate the otherwise valuable contribution of the paper. If addressed, I would be happy to raise my score.

At a high level, these are: a limited experimental section (W1), some unsubstantiated or bold claims (W2), not releasing the simulator (W3), and a deterioration of the writing in the latter parts of the paper (W4).

- W1: The experiment section of the paper (case studies and results) felt quite limited.
  - W1.1: For the ATE case study: the two tasks, while valuable as sanity checks to show the shortcomings of causal models, seemed somewhat constructed and limited. I feel there is a missed opportunity to present an exciting task representative of the real-world manufacturing scenario that the simulator tries to capture. An example (without much knowledge of the topic), could be predicting the effect of an intervention where the parameters of some manufacturing step are modified to see if this has an effect on the quality of the end product. After downloading the dataset, I see such interventions are indeed present, but this was not clear from a first read, which initially made the benchmark and the contribution look very limited. I would recommend adding another task or at least explicitly mentioning that many further experiments are possible with the data you provide (and maybe even offering examples with your code).
  - W1.2: Linear regression outperforms models in the ATE task. This seems like an important outcome from the experiments, particularly from the perspective of a practitioner. However, no discussion is provided, and few details are given to facilitate analysis. Is the regression on all other variables in the dataset? Is the variable indicating the success of the production process a sink node in the graph? If so, this behavior is to be expected in an SCM (i.e., the only variables in its Markov blanket are its parents).
  - W1.3: The results for the causal discovery case studies in section 6.0.2 (Tables 6, 7 and Figure 5) are hidden away in the appendix. This made reading this section somewhat cumbersome and initially made me suspect you had something to hide. If this is due to a lack of space, it would help to state this explicitly.
  - W1.4: Some figures (e.g., 3,4,5) appear to have been made in a rush, e.g., Figure 3 doesn’t list the time units (I guess seconds?), and the x-label should be something like “Causal Discovery Method”; the legend could directly say “Dataset: Small / Medium” instead of “Dataset 1 / Dataset 2”. Or, figure 4 and 5 have inconsistent labels for the y axes, even though they show the same quantity (SHD).
  - W1.5: You mention how the SHD metric is not relevant from large graphs, but still use it as a metric for your results in the causal discovery section. This makes it difficult to get a clearer view of the results to motivate further research and other interesting experiments, which is the unstated goal of a benchmark paper. Showing results for additional metrics [1,2] would help, or at least mentioning that there are alternatives as you talk about the limitations of the SHD.
- W2: Some claims about the datasets are too bold or must be taken at face value. For example:
  - W2.1: In the abstract it says, “we show the nontrivial behavior of a well-known manufacturing process”. Given that there is no real-world data from the actual manufacturing process, I am not sure how you show this. Furthermore, even when it comes to the simulator, I couldn’t find support for this claim, except perhaps the single example about categorical parents in appendix B.1. You describe elements of this complexity in section 4.1 and 4.2, but from the line in the abstract I expected some sort of “evidence”, e.g., some visualization of data (the effect of batching or some non-standard structural assignments). It would also be super helpful to give the explicit expression of some of the structural assignments, e.g., in the appendix.
  - W2.2: Similarly, in line 296 you say “this complex sampling procedure gives rise to rich and heterogeneous datasets”. Since the datasets are the key contribution of the paper, I would have expected some visualization.
  - W2.3: In line 512 you write “our datasets are first of their kind [because we used expert knowledge]”. You are not the first to do this, as other synthetic datasets are also produced with ample domain-expert knowledge (e.g., the DREAM4 challenge [3], or the Neuropathic Pain simulator that you cite in the introduction). I would soften this claim as it's not necessary: you are already making a valuable contribution without it being true.
- W3: By providing two datasets instead of the simulators themselves, the authors limit the potential contribution of this work. I understand that providing user-friendly software comes with complications, and I am happy to consider doing so as out-of-scope. However, sharing at least the simulator code would greatly increase the contribution of the paper, and I would encourage the authors to do this if possible. I apologize if you do provide this and I missed the link to the codebase.
- W4: While the introduction and background are well written, the writing quality progressively deteriorates as the paper goes on. I’ve added some pointers in the “questions” section below.

[1] Peters, J., & Bühlmann, P. (2015). Structural intervention distance for evaluating causal graphs. Neural computation, 27(3), 771-799.
[2] Wahl, Jonas, and Jakob Runge. "Metrics on Markov Equivalence Classes for Evaluating Causal Discovery Algorithms." arXiv preprint arXiv:2402.04952 (2024).
[3] Greenfield A, Madar A, Ostrer H, Bonneau R. DREAM4: Combining genetic and dynamic information to identify biological networks and dynamical models. PLoS One. 2010 Oct 25;5(10):e13397. doi: 10.1371/journal.pone.0013397. PMID: 21049040; PMCID: PMC2963605.

**Questions:**

In no particular order:

1. In related work / datasets and benchmarks: in the paragraph about real-world data, I would also reference [1,2], e.g., after referencing the causal chambers, I would add “Additionally, [1,2] provide real-world datasets with a more or less justified ground-truth causal graph.”
2. In the discussion, under limitations, it would be good to remind the reader that all the usual limitations of synthetic data still apply. For example, the simulators are structural causal models, which means one cannot use this benchmark to evaluate whether these causal models are good models of reality.
3. For the complete ground-truth graphs of appendix F, would it be possible to color the nodes according to observable/hidden variables? This would greatly improve readability and make the different confounding structures visible.
4. Paragraph 412-413: The paragraph offers an explanation as to why causal models are preferable to regression-based techniques. However, the reasoning contradicts the results, since a simple regression drastically outperforms all methods despite having confounders in the data. Could you elaborate on this?
5. Wording / typos
- Line 297: “to a rich and [...] datasets” -> “to rich and [...]” or datasets in singular.
- Line 266: “obtaining” -> “obtained”
- Line 021: “well-known” -> “well-understood” as it’s not like the manufacturing process is “famous” :)
- Line 408: The sentence around “a slightly harder” seems incorrect -> “which is slightly harder” or “a slightly harder task”
- Line 332: section title “Case-Studies” should be “Case Studies” without a hyphen
- Line 394: Section number is 6.0.1 -> shouldn’t this be 6.1 instead?
- Line 1010: “way run” -> “we run?”
- Line 412: “those causal models”. Which causal models are you referring to here? Do you mean “all causal models”.
- Line 468: “Tables 6 and 7 shows” -> “show”
- Lines 258: very minor and pedantic comment: it appears you wrote “i.e.” instead of “i.e.\” in latex and you have a full space. Also, in modern American English these latinisms should be followed by a comma, e.g., “i.e.,”


[1] Mogensen, S. W., Rathsman, K., & Nilsson, P. (2023). Causal discovery in a complex industrial system: A time series benchmark. arXiv preprint arXiv:2310.18654.
[2] Mhalla, L., Chavez-Demoulin, V., & Dupuis, D. J. (2020). Causal mechanism of extreme river discharges in the upper Danube basin network. Journal of the Royal Statistical Society Series C: Applied Statistics, 69(4), 741-764.

---

> ### Author Response · Authors · 2024-11-22
> **Author's response (1/2)**
>
> Thank you for your extensive review and constructive feedback, which helped a lot in improving the manuscript, and for recognizing the effort that was necessary to integrate expert knowledge into the DGP, starting from firstly gathering domain knowledge from experts, to successively encoding it into a DGP based on SCMs.
>
> We first start by addressing your questions/suggestions:
> - Added references to "related work" section.
> - Added to the draft a reminder that the simulator is based on SCMs, therefore inheriting all its underlying assumptions and limitations.
> - Updated figures for the ground truth causal graphs, highlighting in red/orange visible variables, and in light blue the hidden ones. It is interesting to see how some portions of the causal graphs are hidden. The advantage of integrating expert knowledge into the simulator is also that we can model those latent structures, which we know are there exclusively because we have knowledge about the underlying physics.
>
> - We confirm that yes, in the first task the target variable is in the markov blanket of the outcome variable, more precisely a parent. We remark that the gap in performance is only present for the first and easiest Effect estimation task where we are intervening inside the markov blanket of the outcome variable. Therefore, as you wrote, this behavior is expected for SCMs. On the second task, we are not intervening inside the markov blanket anymore, and indeed the gap disappears, making all models perform equally poorly. We added this remark to the main paper. The (deep) causal models we tested, upon further development towards scaling them to real world scenarios, have a bigger potential/margin for improvement. Indeed, on the second task methods based on Normalizing Flows (CAREFL and CNF) are achieving the most promising metrics (Table 3 and 4). Still, no model shows the capacity to leverage bigger amounts of data (See Fig. 2), apart from CAREFL (Slightly). The regression-based techniques, even though performing well on simple tasks, do not show any margin of improvement in complex tasks.
>
> - Corrected the list of typos and wording. Additional ones have been corrected.
>
>
> # W1 to W4
>
> Going on a deeper discussion on W1-W4:
>
> ## **W1: Experimental Section**
>
> We agree that there is a lot of potential, and that we should find the best way to express it. In order to enable researchers to get the most out of our simulator, we adopted the following measures:
> * **We will release the simulator code, enabling users to generate as mush observational and interventional (hard/soft, multiple, and hidden interventions) data as needed.**
> * We are going to add a new and more realistic task. The two tasks that we study in the paper are relatively simple because all causal models are already failing to provide meaningful results, so we did not try to push further with a more complex tasks. From a "stress-test" perspective, that is already enough to show many limitations of the tested models. However, we agree that this might be misleading for the reader, inducing to think that this benchmark is very limited. To address this, we are aiming to provide a more realistic task. Additionally, once the simulator will be publicly available, it will be possible to experiment with arbitrary new tasks.
>
> * W1.2: Yes, in the simplest ATE task we are intervening on the parent of the outcome variable, therefore this behavior is expected in an SCM. We added this clarification to the manuscript in the new revised draft.
>
> * W1.3: Added a clear statement that more results for causal discovery have been moved to the appendix due to lack of space.
>
> * W1.4: We updated the figures, which now have more consistent and bigger labels.
>
> * W1.5: The main concept is that there might be radical differences between different sections of the graph in terms of confounders, causal mechanisms and noise models. In our datasets (and also in other real world scenarios) we have some parts of the graph where almost all variables are observable, and other parts of the graph where variables are mostly hidden. Therefore confounded relationships are not "uniformly distributed", making some parts of the graph easier to discover with respect to others. A similar discussion can be made with respect to structural equations and noise models, as in some parts of the graph causal mechanisms are mostly binary and deterministic (The monitoring structure), while in others they are only nonlinear and noisy. We find your suggestions insightful, and agree that new metrics can give clarity, therefore **We evaluated the discovered causal graphs with SID on CausalMan Small, and added it to the new revised manuscript. We are also trying to implement new metrics from the second paper cited in your review (Code was not release for that paper). Given the size of the causal graphs, the s/c metrics were impossible to evaluate, as they scale exponentially, therefore we are trying to evaluate their approximation. **
>
> (1/2)

---

> ### Author Response · Authors · 2024-11-22
> **Author's response (2/3)**
>
> ## **W2: New Illustrations**
>
> * W2.1: We added a sub-new section in the appendix (C.2) containing a detailed description of the physical models being used by the DGP, along with appropriate references. Furthermore, we added **new illustrative plots** for the monitoring and conditional dependencies.
>
> * W2.2: Added figures 4 and 5.
>
> * W2.3: We agree with your suggestions on softening some claims, therefore:
>     * Softened claim related to the use of expert knowledge for building the simulator.
>     * Softened claim in line 512.
>
>
> ## **W3: Release of DGP**
>
> We are going to release the simulator code, as in the supplementary material there is only the code for running the experiments for effect estimation and causal discovery. With the new code it will be possible to generate new observational and interventional data (hard/soft, multiple, and hidden interventions).
>
> Moreover, as written above, we tried to make DGP as clear as possible by adding a new section in the appendix C.2 where we describe all the causal mechanisms involved. Additionally, we have two new visualizations (Figure 4 and 5).
>
> ## **W4: Writing Quality**
>
> After reading the second part of the paper, we agree that the writing quality on the second part of the paper is definitely less polished. Apart from implementing all the typos you indicated, we corrected a few additional ones.
>
> ## Corrections
>
> A recap of the corrections we implemented:
>
> - New section C.2 with  description about structural equations.
> - New Illustrations (Figure 4 and 5).
> - Polished figures.
> - Polished writing.
> - Added references.
> - Added SID for Causal Discovery on Causalman Small.
>
> As soon as we have the results for the new Effect Estimation task, we will add it to the appendix.
>
> We thank again for the review. Fee free to ask if further information is needed, or if you have any additional feedback,

---

> > ### Comment · Reviewer_PVqx · 2024-11-26
> >
> > I thank the authors for their work on the updated manuscript and for addressing my concerns.
> >
> > Particularly, I find the new and updated plots and sections in the appendix have added significantly to the value of the work.
> >
> > Therefore, I recommend acceptance and raise my score trusting that the authors will release the simulator code and a new task.

---

> ### Author Response · Authors · 2024-11-26
>
> We would like to express our gratitude for your constructive review and for increasing your score.
>
> **We added the results for the new task tp the manuscript**. Indeed, the new results indeed show that on nontrivial tasks where confounders and more nonstandard causal mechanisms are present, linear regression is definitely the worst method overall, therefore supporting our statements in the main text.  Other methods, instead, are not degrading as much as linear regression. We added the new results in the appendix (Fig. 8 and Table 4), and added remarks and references about it in the main text.
>
> In this new task, we are intervening on the variable PF_M1_T1_Force, which describes the press-fitting force. The treatment is setting this force to an extreme value. Additionally, this variable has multiple bidirected edges with other variables describing the PF process, and it is also a direct parent of other variables. Therefore an extreme intervention can cause anomalies to propagate through the graph, influencing other physical quantities that depend on it (For example s_grad and F_max).
>
> Additionally, even if we did not cover Root-Cause Analysis in this paper, we note how this simulator can model the propagation of production anomalies though the causal graph, therefore extending its applicability to RCA and Causal Explanations ("Why did this anomaly happen, and which are the responsible nodes for it?").
>
> Lastly, we also added a new metric for causal discovery, namely the p-SD where we are counting the number of different separation statements (where S = Parents X Union Parents Y) between the two ground truth and discovered causal graphs. For the SID, it was possible to evaluate it only on the smaller dataset due to computational reasons.
>
> ## Corrections
>
> * Added new task.
> * Improved the writing quality in the second part.
> * Added SID and p-SD for causal discovery.
>
>
> Thank you again. Feel free to ask if any additional suggestions or information are needed.

---

### Official Review · Reviewer_y52m · 2024-11-01

**Soundness:** 1
**Presentation:** 1
**Contribution:** 1
**Rating:** 3
**Confidence:** 4

**Summary:**

This paper presents a new benchmarking dataset for causal inference and discovery that is meant to mimic a manufacturing process. The simulator is “hand-tuned” by “domain experts” with the goal of obtaining a benchmarking dataset that is more realistic that existing benchmarking results.

**Strengths:**

The paper studies the challenge of benchmarking in the field of causality, which is an important problem that benefits from more work and proposals for datasets.

**Weaknesses:**

This paper is not of high quality, has no clear (scientific) contribution, is not reproducible, and is not written well.
First, the proposed data-generating process is not described properly. There are no equations that formalize the system that is simulated, which makes the proposed benchmark not reproducible and impossible to analyze for researchers. Moreover, none of the simulation choices are justified scientifically based on any evidence beyond the claim that “domain experts” were involved in designing the data-generating process. The work provides no evidence that their released dataset is “realistic”, in whatever interpretation of “realistic” on may consider, compared to the existing benchmarking datasets referenced in Section 2. Specifically, the work does not justify what exactly is wrong with existing synthetic and semisynthetic benchmarking settings and in what we the proposed dataset resolves it (e.g., type of causal mechanisms, variance artefacts, signal to noise ratio, causal structure, etc). Independent of these points, the writing is not polished, and the font in most figures is unreadable.

Overall, the paper does not motivate a concrete gap in the field that it seeks to address, it does not justify why the proposed dataset is superior for benchmarking, and it does not provide an accessible way to understand the data-generating process that is proposed. I recommend that these questions are thoroughly scientifically analyzed and thought through before considering resubmitting the work.

**Questions:**

n/a

---

> ### Author Response · Authors · 2024-11-19
> **Reproducibility and DGP**
>
> Thank you for your review. We understand your feedback and adopted appropriate measures, especially regarding the Data Generating Process (DGP).
>
> ## Reproducibility and realism of the Data Generating Process
>
> We built a simulator **based on physical models derived from first principles**. To concretely address your review, we added a new subsection in Appendix C where we describe in detail the causal mechanisms involved, including citations to the relevant literature from the field for how those models can be analytically derived.
>
> Additionally, **we will release the code for the simulator itself**. Upon publication, It will be possible to generate observational and interventional data, including soft/hard interventions, multiple interventions, and interventions on latent variables.
>
> To clarify our contribution, we list the main characteristics of this simulator and the research gap that we are trying to address:
> * **Higher level of datatype heterogeneity:** Continuous, binary and categoricals are all present and play important roles in the DGP.
> * **Many Causal assumptions do not hold:** Our DGP does not satisfy the typical causal assumptions such as Additive noise, linearity, or causal sufficiency. Identifiability likely does not hold anymore.
>  * **Causal Mechanisms:** Our simulator can offer causal mechanisms which are uncommon with respect to other publicly available DPGs. Among those, we model also **Conditional Dependences** (Sec. 4.1 and Appendix C.1), which can be seen as second-order uncertainties. Those kind of causal mechanisms result in multimodal and highly asymmetric marginal node distributions. Additionally, noise is often treated nonlinearly.
> * **Focus on Causal Insufficiency**: Although possible to marginalise out variables in any DGP (when perfect knowledge is available), rarely benchmarks have a special focus on causal insufficiency, which can appear in real-world scenarios due to physical, economical or confidential reasons. Moreover, the choice of which variables are hidden and which ones are visible is given by the knowledge of the real production plants.
> * **Availability of Interventional data**: We are going to release the code to arbitrarily generate as much interventional data as needed.
>
>
> For what concerns the *reproducibility of experiments*, instead, we adopted all the necessary measures that are listed in the Reproducibility statement at the end of the main text. In the supplementary material you can find an automatic and configurable framework which can be used to run any of the presented experiments. The only limitation from that point of view is related to one of the core takeaways of the paper, that  massive computational resources are needed for causal models such as Neural Causal Models or Causal Bayesian Networks.
>
> ## Revisions
> Finally, the following revisions have been implemented in the new draft:
> * Add section to the appendix (C.2) describing Causal Mechanisms. Further, we clarified in the main paper that our simulator is based on physical models.
>  * Clarified our contribution and specified what is the research gap that we are addressing.
>  * Correction of typos.
>  * Updated most figures in order to make them more readable. Added new figures for illustrating different aspects of the DGP.
>
>
> Lastly, we started the procedure to release the code for the simulator, and will make it public as soon as possible on a public anonymized repository. By doing so, we enable users to sample as much observational/interventional data as needed, including soft and/or multiple interventions. Interventions on latent variables will also be possible.

---

> > ### Comment · Reviewer_y52m · 2024-11-22
> >
> > Thank you for your comments. My rating remains unchanged.
> >
> > My view stands that the main part of the paper does not properly motivate why this specific simulator is realistic and why we should believe that it is better suited for causal discovery benchmarking than other tools. I am lacking any concrete evidence—qualitative and quantitative—for why this is the case. I do not find the appendix the right place to explain how the simulator works, it should be clear from reading the main paper. Also, the new appendix C is not referenced anywhere in the main text. The few textual changes made in this revision do not change anything significant about the presentation or contribution of the work in the main text.

---

> > > ### Author Response · Authors · 2024-11-28
> > >
> > > Thank you for your response,
> > >
> > > We understand that claiming the realism of our simulator is a strong statement. To address this, we moved a paragraph from the appendix to the main text, at beginning of Sec.4. In this new paragraph we provide a qualitative description of the system that we are modeling, hoping that this will enable readers to intuitively understand the context of the simulator **just by reading the main paper**. Moreover, thanks for making us notice the missing reference to the appendix, which we promptly added in the new revision. Additionally, apart from highlighting the main characteristics of our datasets in the rest of Sec.4, in the related work section we make further remarks on the differences between our simulator and the others.
> > >
> > >
> > > We remark that, although the simulator is one of the key contributions, we envisioned our paper to also concentrate on the results from our experiments. Having a focus on performing experiments permits us to show serious limitations in current causal models, while it also **provides a motivation for why it is exciting to work with this data**. Releasing a simulator without any task to solve might have been dispersive, therefore we did construct different targeted tasks for which all models will be compared.
> > > As we need to stay within the page limit of a conference paper, we are forced to keep the description of the DGP compact, therefore further digressions are kept for appendix C, although, as we wrote above, we still added a new paragraph to the main paper.
> > >
> > > ## Revisions
> > >
> > > Here we list the new revisions:
> > > * Added new paragraph on the main text, at beginning of Sec.4
> > > * Improved writing overall.
> > > * Added missing reference to appendix.
> > > * Remarks on related work section.
> > >
> > > The deadline for revisions is now over. In any case, feel free to make additional suggestions, observations, or to ask for new clarifications.
> > >
> > > Best regards.

---

> > > ### Author Response · Authors · 2024-12-02
> > >
> > > Dear reviewer,
> > >
> > > We have ~27 hours left for answering your messages, and we are still available to address any of your feedback or to provide any additional clarification, if needed. Moreover, a summary of the revisions has also been posted as a general comment.
> > >
> > > In case you think that your feedback was thoroughly addressed, we kindly ask if you could reconsider your score.

---

### Official Review · Reviewer_XZnD · 2024-11-04

**Soundness:** 3
**Presentation:** 3
**Contribution:** 2
**Rating:** 6
**Confidence:** 3

**Summary:**

Real-world causal phenomena often diverge significantly from common assumptions such no latent confounder. These disparities pose substantial challenges to create fair benchmarks due to the scarcity of realistic data with a known data-generating process. This paper presents an extensive experimental evaluation using two large datasets that simulate a real manufacturing scenario with a physics-based model validated by experts in the field. This findings reveal significant performance and scalability limitations in many state-of-the-art causal models, highlighting wide variability in runtime and memory demands.

**Strengths:**

1. This paper addresses one of the most critical challenges in causality: benchmarking different causal models with a particular focus on the scalability and applicability.
2. Overall, the paper is well-structured and clearly written.
3. The experiments are extensive, covering seven causal discovery algorithms and five evaluation metrics.

**Weaknesses:**

1. Why not consider and model the measurement error during CAUSALMAN simulation?
2. How to obtain the ground-truth of two CAUSALMAN datasets? Simply by domain expert? Could you please provide more details on the validation process used by the domain experts or any quantitative measures used to ensure the accuracy of the ground truth?
3. According to your experiments, if you were to recommend one most reliable causal discovery method for real-world datasets, what would it be? Based on your experiments, what are the key trade-offs between the most promising causal discovery methods for real-world datasets in terms of accuracy, scalability, and robustness to different types of data?

**Questions:**

(See above)

---

> ### Author Response · Authors · 2024-11-19
>
> Thank you for your review and your feedback. We address below your points:
>
> **Measurement Error:**
> No, we do not model directly the measurement error in the causal graph, as we are modeling quantities which can be measured accurately, making the measurement error negligible in practice. Additionally, the probability of failure for those sensors is also typically low, if proper maintenance is done.
> If we were modeling other production lines with different underlying physical mechanisms where accurate sensors are not available, the answer could have been different.  As a practical example: When dealing with product testing such as pressure tests, leakage tests, or similar, modeling the measurement error is of crucial importance.
>
> **Ground truth and Data Generating Process:**
> Our simulation has been *built starting from physically justified structural equations that are derivable from first principles*. Therefore, the ground truth has been assembled by integrating expert knowledge about all the physics underlying the production process. **We added appendix C.2 where we make an extensive description of the physical models**, including citations to the relevant literature. This approach is for sure time consuming, as it required to write down every structural equation, although there are repetitions. When modeling physical systems, we remark that this is often the only way possible, as some quantities may be hard or even impossible to measure for physical/economical/confidential reasons.  In appendix Sec.G we can see from the ground truth causal graphs how some causal structures can be present while being completely latent.
> **Moreover, we will release the code containing the DGP, which will enable researchers to generate as data as needed, including interventional data (including hard/soft interventions, multiple interventions and interventions on hidden variables).**
>
> **Best Causal Discovery algorithm:** From our tests we see a significant trade-off, where the best performing methods on the "small" scale can be among the worst on the large scale. **The PC algorithm, indeed, showed the most robust behavior among the tested methods on CausalMan Small**, especially in terms of precision and recall.
> Additionally, as in real world data, our data exhibits node distributions which can be asymmetric and multi-modal due to conditional dependencies (See Appendix C.1), which favors constraint-based methods.
>
> Moving to CausalMan medium, we see that ML-based methods are advantaged overall, both in terms of performance but also runtime. However, even the best performing methods are still very distant from providing satisfactory results that can be used in any real downstream tasks. Therefore we consider Causal Discovery at the large scale still an entirely open research problem.
> We also remark that the amount of data is an important factor, as we can see in Fig. 6 in the appendix. In that figure, we can see that the PC algorithm is strongly disadvantaged in the big-data regime.
>
> NOTE: Data was normalized in every experiment, making it hard for continuous optimization-based methods to exploit patterns of increasing marginal variance (See the concept of varsortability in "Beware of the Simulated DAG! Causal Discovery Benchmarks May Be Easy To Game" from Reisach et al.2021).
>
> Thank you again for your review, we updated our manuscript with a description of the DGP. Please feel free to ask if any additional clarifications are needed or if there are further suggestions.

---

> > ### Comment · Reviewer_XZnD · 2024-11-20
> >
> > Thank you for your responses, which have satisfactorily addressed my concerns. In particular, the details provided about the data-generating process significantly enhance the reproducibility and utility of the work, allowing readers to generate data as needed, thereby broadening the impact of this contribution.
> >
> > Overall, I find that this paper tackles an important problem—benchmarking practical and large-scale causal discovery. Given its contributions, I am inclined to recommend its acceptance and have updated my score to 6.

---

> > > ### Author Response · Authors · 2024-11-22
> > >
> > > Thank you again for the valuable feedback and for increasing your score. We think that having this simulator will enable researchers to explore deeper large-scale causality in practical real world scenario, and that this could stimulate the development of methods and models with better performance and compute requirements.
> > >
> > > As written before, we are available at any moment for further information or suggestions.

---

### Official Review · Reviewer_1EgP · 2024-11-04

**Soundness:** 2
**Presentation:** 2
**Contribution:** 2
**Rating:** 5
**Confidence:** 3

**Summary:**

This paper presents a critical evaluation of the scalability and effectiveness of current causal models through extensive empirical analysis on two novel large-scale manufacturing datasets. The authors demonstrate significant limitations in both causal inference and discovery methods when applied to realistic, complex scenarios. The work provides valuable insights into the practical challenges of applying causal models to real-world problems and releases new benchmark dataset for research in high-dimensional causality.

**Strengths:**

The paper addresses real-world manufacturing scenarios and incorporates domain knowledge thus having high practical relevance.

There's a rigorous empirical evaluation with multiple experimental settings and ablation studies.

**Weaknesses:**

The dataset is very specific which while valuable there should be more domains considered (e.g. biological networks that have different DGP distributions).

Mixing two evaluations in one. I feel that this could be two papers each expanding on different sides of causal modeling. One for scalability of causal discovery methods and one for causal inference. This would have the room to expand on insights and evaluations.

Odd result regarding linear regression (check questions).

**Questions:**

Missing related work:
https://arxiv.org/abs/2406.03209 studies bayesian causal discovery methods.

Why are there both logistic and linear regression as baselines? I understand that the outcome is a binary variable so why use linear regression? Also, for logistic regression, is MSE the appropriate metric?

Also, I would like to understand the linear regression performance a bit better. What is the groundtruth DGP considered? What is the non-linearity involved? I think it's possible to construct a non-linear SCM for which linear regression would fail and many applications will be more non-linear than linear. Can you provide plots of what the models learned in each case vs what's the groundtruth function (or samples)?

Regarding CausalMan. How do the SCMs proposed here better than other simulators out there (e.g. https://gnw.sourceforge.net/dreamchallenge.html for gene regulatory networks)? I understand this is a different domain but why a different domain makes it better? How do you evaluate the quality of the SCMs you propose? I would like to understand why this is a great benchmark and the paper doesn't provide sufficient evidence.

In table 2. the MMD entry for CNF is Nan. Is this by accident?

---

> ### Author Response · Authors · 2024-11-20
> **Author response (1/2)**
>
> Thank you for your review, we appreciate your feedback and observations.
>
> First of all:
> * We added the missing citations. Thanks for pointing out to this new reference.
> * In table 2 the entry for MMD for Causal Normalizing Flows is Nan, but not by accident: When scaling those models, training instabilities did often appear. At sampling time, the model did uniformly output more than one variable with diverging values (>>1e14), yielding the MMD computation to output Nan.
>
> Regarding the DGP and the performance gap for linear regression will be instead addressed in the section below.
>
> ## Data Generating Process
>
> Our simulation has been built starting from physically justified structural equations that are derivable from first principles. Therefore, the ground truth has been assembled by integrating expert knowledge about all the physics underlying the production process.
>  **For a detailed explanation of every structural equation, including illustrations about the main mechanisms, we added Sec. C.2 in the appendix where we describe the underlying DGP, with citations to the relevant literature. Moreover, the code to generate new observational/interventional data will be released.** We hope that all this new information will sustain our claims about our results, and about the realism of our simulator. Additionally, we highlighted more concretely our contribution in the "contribution" and "related work" sections.
>
> In connection with the performance of linear and logistic regressions, We start by remarking that linear regression is competitive only on the first and simplest task, and degrades to a level comparable to other methods in the second task.
> We did choose to keep the linear regression because it is the simplest and most interpretable baseline possible, even if it does not reflect many of the assumptions of the underlying DGP. Moreover, it is a common baseline against which causal models are often tested.
>
> Exploring deeper its performance on the first task, we start by remarking that target and outcome variables are both binary, and the structural equation is a logic AND between all the (binary) parents, which is a nonlinear causal mechanism. Moreover, we are intervening on a parent node, which is on the markov blanket of the target variable. Therefore, this behavior is expected when working on data generated from a SCM, as we are modeling a direct causal relationship.
>
> For the second task, both variables are still binary, but the target is now a
> grandparent of the outcome node, which is not in its markov blanket anymore. In
> this task, we can already see in Table 3 and 4 that the linear regression does not have any advantage anymore.
>
> Lastly, to address your question regarding the use of MSE for logistic regression, we remark that we used the model only for treatment effect estimation, and not for estimating interventional distributions. Therefore we used the MSE to measure the distance between the estimated ATE and the ground truth ATE, which are both scalars.
>
> (1/2)

---

> ### Author Response · Authors · 2024-11-20
> **Author's response (2/2)**
>
> ## Contribution of our simulator
>
> As written above, in order to sustain our statements about the realism of the DGP, we added a new section in the appendix (C.2), where we justify at a physical level the underlying DGP. We will highlight also here our main contributions and research gap that we want to address with our simulator and benchmark:
>
> * **Higher level of datatype heterogeneity:** Continuous, binary and cate
> egoricals are all present and play important roles in the DGP.
>
> * **Many Causal assumptions do not hold:** Our DGP does not satisfy
> the typical causal assumptions such as Additive noise, linearity, or causal
> sufficiency. Identifiability likely does not hold anymore.
>
> * **Causal Mechanisms:** Our simulator can offer causal mechanisms which
> are uncommon with respect to other publicly available DPGs. Among
> those, we model also Conditional Dependencies (Sec. 4.1 and Appendix C.1), which
> can be seen as second-order uncertainties. Those kind of
> causal mechanisms result in multimodal and highly asymmetric marginal
> node distributions. Additionally, noise is often treated nonlinearly.
>
> * **Focus on Causal Insufficiency:** Although possible to marginalise out
> variables in any DGP (when perfect knowledge is available), rarely benchmarks have a special focus on causal insufficiency, which can appear in
> real-world scenarios due to physical, economical or confidential reasons.
> Moreover, the choice of which variables are hidden and which ones are
> visible is given by the knowledge of the real production plants.
>
> * **Availability of Interventional data:** We are going to release the code
> to arbitrarily generate as much observational and interventional data as needed. It will be possible to generate hard/soft interventions, multiple interventions, and interventions on latent variables.
>
> ## Corrections:
>
> The following corrections have been implemented in order to address your feedback:
> * Added new section in Appendix C.2, where we describe all causal mechanisms involved, plus illustrations (Fig 4 and 5) about the most characteristic ones.
> * Our contribution has been written more precisely in the "contribution" paragraph in the introduction.
> * Specified why on the first task we are getting those results for linear regression.
> * Added missing reference.
>
> Lastly, *we will add the requested visualization in the appendix in the next revision* and, as suggested by reviewer PVqx, we are working towards adding a new effect estimation task. Moreover, we remark again that the code for the simulator will be made available, allowing researchers to generate new observational and interventional data, including hard/soft interventions, multiple interventions, and interventions on latent variables.
>
> Thank you again for your review. Please feel free to ask if any additional clarifications are needed or if there are further suggestions.
> (2/2)

---

> ### Author Response · Authors · 2024-11-27
> **New visualization for interventional distributions**
>
> Dear reviewer,
>
> We follow up on our past message: To present a different perspective on why a simple linear regression is outperforming all other methods (only) in the first ATE task, **we added a new figure( Fig.11)**. Now we can observe the interventional distributions that the deep causal models (+ Causal Bayesian Networks) have learned. Similarly to what is suggested in Fig. 2b, we can see also in Fig.11 that these models are typically not consistent in predicting interventional distributions, and hardly a model can estimate accurately BOTH treated and control population. This leads to a decrease of ATE performance.
>
> We think that results are much more reasonable if we also take into account the performance in the **new third ATE task** suggested by reviewer PVqx. In this new task, we are intervening on the node PF_M1_T1_Force. This variable has multiple confounded relationships with other variables describing the PF process, and it is also a direct parent of other variables. Furthermore, the path from the target to the outcome variable now has multiple nodes in between. What we can concretely observe is that, on a harder task, results are more consistent with what would be expected from those models. **In this new task, indeed, linear regression is clearly disadvantaged, and other models instead perform much better at modelling nonlinearities and confounding effects.** See Fig.10 and Table 4 in the appendix for the new results and plots.
>
> We hope that this clarified further your questions, and feel free to ask for any additional clarification.

---

> ### Author Response · Authors · 2024-12-02
>
> Dear reviewer,
>
> We have ~27 hours left for answering your messages, and we are still available to address any of your feedback or to provide any additional clarification, if needed. Moreover, a summary of the revisions has also been posted as a general comment.
>
> In case you think that your feedback was thoroughly addressed, we kindly ask if you could reconsider your score.

---

> > ### Comment · Reviewer_1EgP · 2024-12-02
> >
> > Dear authors, I appreciate the long and thoughtful responses.
> >
> > However I still believe that the evaluation is limited. In a way, the paper is doing the following: you suggest that evaluation of causal models is challenging, then you are proposing a new evaluation and you conclude that the evaluation of causal models is challenging. This is a circular argument and the paper doesn't provide a convincing story why this new evalaution should be accepted by the community. To be clear, I see value in new benchmarks but if the topic of the paper is introducing and discussing benchmarks, I would expect a more concrete methodology - a very challenging task (how one compares the benchmarks?) but one that needs to be done.
> >
> > For the reasons above, i'm keeping my score.

---

> ### Author Response · Authors · 2024-12-03
>
> Dear reviewer,
>
> Thank you for your answer.
>
> We agree with you that designing a benchmark is always challenging, as it requires a validation procedure which is not trivial. To explain our methodology, we think it is necessary to also articulate our requirements. We built this simulator with the expectation to compensate some areas where other simulators and datasets were not meeting our needs. What breaks the circular argument is that **the benchmark was not intentionally created to show how limited current methods are**. The simulator was built to meet a list of requirements which we think are missing in other benchmarks. From those requirements, we did create a simulator. What we show in the paper is that current methods do not perform well when we deal with data having those characteristics.
>
> In other benchmarks, what we found limiting was the scarce amount of datatype heterogeneity, the type of causal mechanisms (and noise models), and the focus on causal sufficiency. To answer your doubts about our methodology, we will focus on the last point.
>
>
> Many simulators ignore that the majority of variables are latent, and they often try to build a "causally sufficient" version of the real data, discarding all external confounding effects. In our case, we needed a grounded way to model what happens on hidden nodes, and **how** they influence observable ones. In the simulator, some parts of the causal graph are there **only because we know the underlying physics**, but they are never observable in the real world. We managed to do this by integration of expert knowledge. For example, the computation of the effective elasticity of a Magnetic Valve is hidden, but we can still manage to model it in the SCM by deriving the underlying physical model using the theory of elasticity. In this scenario, a quantitative estimate is extremely hard -if not impossible- to carry on without heavy (and unrealistic) assumption on the DGP. We want to compensate this limitation by using physical models derived from first principles (Sec. C.2).
>
> Indeed, to bypass this step, we pursued what is typically considered standard practice in numerical mathematics, physics, and simulations. For example, in Computational Fluid Dynamics, simulations are mostly performed starting from some system of ordinary/partial differential equations (Navier-Stokes, Laplace equation, et similia) derived from the fundamental laws of dynamics **which we assume to be true**, and by successively applying some form of discretization.
>
> Therefore, what we might conclude is that we used a methodology which is not common in the Machine Learning field, as it is less data-driven than other approaches, but which we still consider concrete and highly based on our fundamental knowledge of the system we are modeling.

---

### Author Response · Authors · 2024-11-28
**Thanks to reviewers, additional remarks and general summary**

We sincerely thank all reviewers for the detailed and valuable feedback. Your constructive criticism helped us to greatly improve our manuscript.

## General overview

In general, the main focus of your feedback rightly focused on the key contribution of the paper, the Data Generating Process. To address it in the most direct way possible, we added a new paragraph in the main paper (beginning of Sec.4), and extended Sec. C in the appendix, where in the latter we provide an explanation for each Structural Equation and its relative physical intuition. We also added new illustrations for special mechanisms such as for the monitoring system and for conditional dependencies. Moreover, we also remark our focus on heterogeneous data-types, non-additive and nonlinear noise modeling, and causal insufficiency. Importantly, **we will release the source code needed to generate new observational and interventional data (hard/soft, multiple and hidden interventions)**. With the simulator available, the potential of this simulator can even extend beyond the scope of this paper, as it constitutes also a good test-bench for Root-Cause Analysis, Causal Explanations, and more, which is one of our future goals. **Upon publication, we will create a public website, where this benchmark will be presented as a challenge, and where we will keep a leader-board with the best performing methods.**

Secondly, another point shared by more than one reviewer was the performance on the ATE tasks, which were not described with enough depth. Therefore, we added different remarks and descriptions on those results. Importantly, **we remark that linear regression performed well only on the first ATE task** (See Fig. 10 for illustration about what causal models learned). To further support our statements, we also added a third ATE task where we are intervening on a part of the graph where causal mechanisms are nonlinear and highly confounded. The new results are now much more consistent with what is expected, as the linear regression is clearly disadvantaged when the task is not trivial anymore, whereas causal models show stronger robustness.


## Summary of Revisions

Lastly, here we present a general overview of the additions and revisions that were made to the manuscript:

* Polishing of the writing and figures, which now are much more readable.
* New paragraph in Sec.4 describing qualitatively the system that is being modeled.
* Extended section C.2, describing the Structural Equations and their relative physical intuition.
* Added a third ATE task involving more nonlinearities and confounding effects.
* New illustrations for the most characteristic causal mechanisms such as monitoring and conditional dependencies.
* New illustration for what models learned in first ATE task.
* Polished illustrations about the ground truth causal graphs: Visible and latent variables have now different colors.
* Added new references and remarks when requested.

The deadline for revisions is now passed. However, feel free to ask for any additional information or clarification.

Thank you again and best regards.

---

### Meta-Review · Area_Chair_dcz2 · 2024-12-23

**Metareview:**

This paper presents two large-scale manufacturing datasets, derived from a physics-based simulator with domain-expert input, and extensive evaluations of various causal models. On the positive side, reviewers see potential value in offering benchmark data that reflect realistic conditions (e.g., high dimensionality, hidden confounders, and nonlinear relationships). They also appreciate the authors’ thorough empirical tests on runtime, scalability, and performance gaps across methods. However, concerns persist around the paper’s clarity in describing the simulator’s data-generating process, the justification for its “realistic” nature, and reproducibility details.

**Additional Comments On Reviewer Discussion:**

All reviewers participated in the discussion and acknowledged the rebuttal.

---

### Decision · Program_Chairs · 2025-01-22

Reject